# Cryo-EM structure of a RAS/RAF recruitment complex

Eunyoung Park[1,2,4,6], Shaun Rawson ●[2,6], Anna Schmoker ●[1], Byeong-Won Kim ●[1,5], Sehee Oh[1], Kangkang Song[3], Hyesung Jeon ●[1,2] ✉ & Michael J. Eck ●[1,2] ✉

RAF-family kinases are activated by recruitment to the plasma membrane by GTP-bound RAS, whereupon they initiate signaling through the MAP kinase cascade. Prior structural studies of KRAS with RAF have focused on the isolated RAS-binding and cysteine-rich domains of RAF (RBD and CRD, respectively), which interact directly with RAS. Here we describe cryo-EM structures of a KRAS bound to intact BRAF in an autoinhibited state with MEK1 and a 14-3-3 dimer. Analysis of this KRAS/BRAF/MEK1/14-3-3 complex reveals KRAS bound to the RAS-binding domain of BRAF, captured in two orientations. Core autoinhibitory interactions in the complex are unperturbed by binding of KRAS and in vitro activation studies confirm that KRAS binding is insufficient to activate BRAF, absent membrane recruitment. These structures illustrate the separability of binding and activation of BRAF by RAS and suggest stabilization of this pre-activation intermediate as an alternative therapeutic strategy to blocking binding of KRAS.

RAF-family kinases are a central point of control in the signaling apparatus that regulates cellular proliferation, growth, and differentiation[1,2]. RAFs are maintained in an inactive, autoinhibited state in the cytosol and are activated downstream of receptor tyrosine kinases via recruitment to the plasma membrane by GTP-bound RAS. Upon activation, RAFs phosphorylate their sole known substrate MEK1/2, and activated MEK in turn phosphorylates ERK1/2. Somatic mutations in RAF can subvert the normal RAS-dependent activation process and are a frequent cause of cancer[3,4]. In particular, the V600E mutation in BRAF drives more than half of malignant melanoma and is also found in lung, colorectal, thyroid, and many other cancers[5]. The conserved domain structure of BRAF and other family members (ARAF and CRAF, also called RAF1) includes the RAS-binding domain (RBD), a cysteine-rich domain (CRD), and the C-terminal kinase domain (Fig. 1a). The RBD and CRD lie in the N-terminal regulatory region of the protein, which is crucial both for maintaining autoinhibition and for RAS-driven activation and membrane association[1,2]. We and others have shown how BRAF

is locked in an inactive state as a complex with its substrate MEK and a 14-3-3 dimer[6,7]. In this autoinhibited configuration, the 14-3-3 dimer binds serine phosphorylation sites that flank the BRAF kinase domain (pS365 and pS729), allowing it to restrain both the BRAF kinase and CRD domains in a cradle-like structure that precludes BRAF dimerization, which is crucial for its activation[8]. The CRD domain lies at the center of the autoinhibited complex, where it is largely protected from interactions with the membrane and RAS. The RBD, by contrast, is exposed and available to engage RAS. In the active state, driven by RAS engagement at the plasma membrane, these components reorganize such that the 14-3-3 domain binds the C-terminal pS729 site of two BRAF proteins, driving the formation of an active, back-to-back BRAF kinase dimer[6,7,9,10]. The N-terminal regulatory region appears not to engage with the dimeric kinase/14-3-3 module[6,9]; instead, it is thought to bind RAS at the membrane[11–13].

Structural studies of RAS bound to RAFs have focused primarily on the isolated RAF RBD domain[14,15], and more recently on a fragment

[1]Department of Cancer Biology, Dana-Farber Cancer Institute, Boston, MA 02215, USA. [2]Department of Biological Chemistry and Molecular Pharmacology, Harvard Medical School, Boston, MA 02115, USA. [3]Department of Biochemistry & Molecular Biotechnology, University of Massachusetts Chan Medical School, 364 Plantation St, Worcester, MA 01605, USA. [4]Present address: Pfizer R&D Center, 3200 Walnut St, Boulder, CO 80301, USA. [5]Present address: New Drug Development Center, Osong Medical Innovation Foundation, Cheongju, Chungbuk 28160, Republic of Korea. [6]These authors contributed equally: Eunyoung Park, Shaun Rawson. ✉e-mail: hjeon@crystal.harvard.edu; eck@crystal.harvard.edu

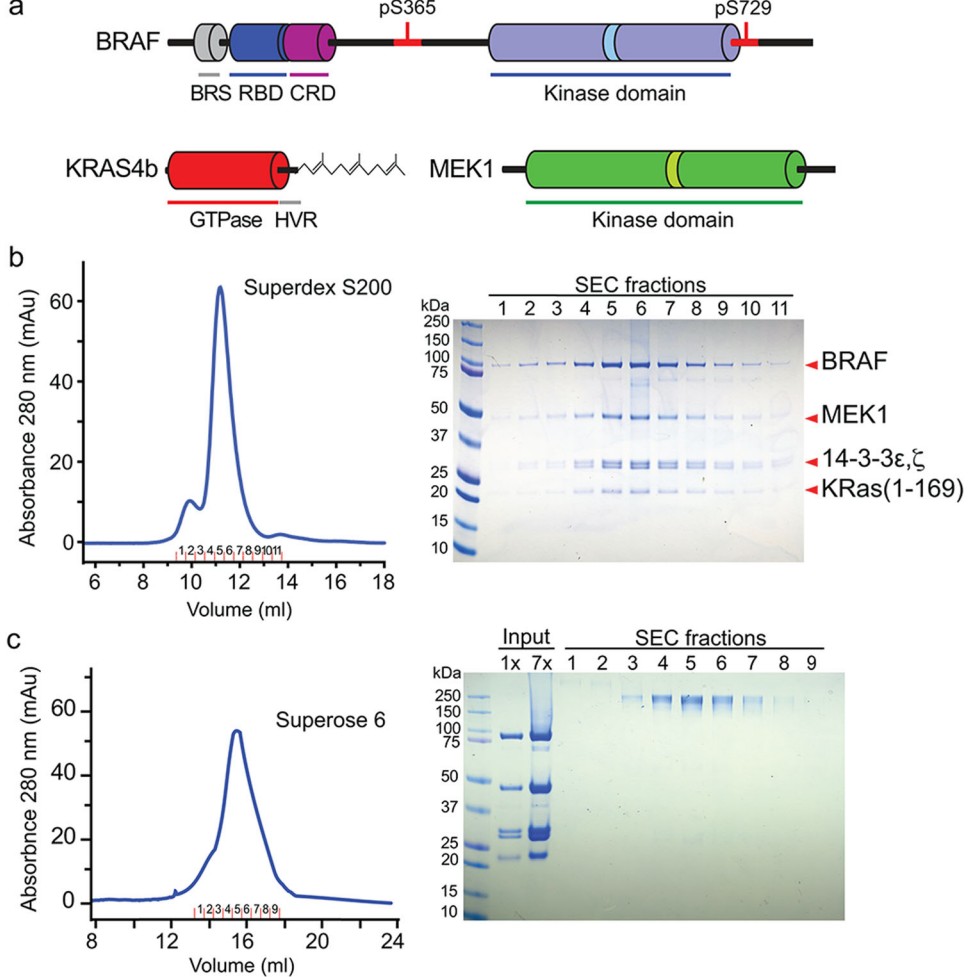

**Fig. 1 | Preparation of KRAS/BRAF complexes for structural analysis.**
**a** Schematic of domain structures of BRAF, KRAS, and MEK1. Key phosphorylation sites are indicated above the schematics. Binding sites for the 14-3-3 domain in BRAF are indicated in red. BRS BRAF-specific domain, RBD RAS-binding domain, CRD cysteine-rich domain, HVR RAS hypervariable region. **b** Size-exclusion chromatography of the KRAS/BRAF/MEK1/14-3-3 complex. Elution profile of the complex on Superdex S200 is shown on the left, and a Coomassie-stained SDS-PAGE gel of the indicated fractions is shown on the right. The experiment was performed more than three times with similar results. **c** Size-exclusion chromatography of a BS3-cross-linked KRAS/BRAF/MEK1/14-3-3 sample. The complex analyzed in (**b**) was subjected to cross-linking with BS3 and re-examined with size-exclusion chromatography. Elution profile of the complex on Superose 6 is shown on the left, and a Coomassie-stained SDS-PAGE gel of the indicated fractions is shown on the right. Aliquots of the input sample before and after approximately sevenfold concentration, but prior to cross-linking, are also shown on the gel. The experiment was performed three times with similar results.

of CRAF containing both the RBD and CRD domains[16–18]. These studies have provided a detailed view of intermolecular interactions between these domains, along with insights into how they may assemble in a membrane context[16]. In brief, RAS proteins engage the RBD domain largely via their switch I region, including an extensive β-strand interaction with the β2-strand of the RBD. RAS also engages the adjacent CRD domain, although the interaction apparently contributes little to binding affinity. The CRD domain therefore has a dual role in the active state, in that it binds both the membrane and RAS.

Despite this detailed understanding of KRAS/RAF recognition, it remains unclear how this interaction results in RAF activation. The binding of the KRAS GTPase domain alone is not sufficient to activate RAF; KRAS must be prenylated at its carboxy terminus (farnesylated or geranylgeranylated) in order to mediate cellular transformation[19]. Prenylation promotes association of KRAS with the plasma membrane, and membrane association is crucial for activation of RAF. Indeed, grafting of the 17 C-terminal residues of KRAS onto CRAF, including a lysine-rich segment and the prenylation signal (a C-terminal CaaX motif), is sufficient to promote its membrane localization and activation in a RAS-independent manner[20,21].

A recent cryo-EM structure of the BRAF/MEK1/14-3-3 complex revealed a fixed, rather than flexible, RBD domain with defined interactions with the 14-3-3 domain[7]. Structural modeling based on this observation and on structures of RBD-CRD complexes with RAS led to the suggestion that upon recruitment of BRAF to the membrane by KRAS, binding of KRAS to the autoinhibited BRAF complex could directly promote RAF activation by sterically perturbing the inhibitory interactions of both the RBD and CRD domains and freeing the CRD domain to interact with KRAS and the membrane[7]. In an alternate model, engagement with RAS does not directly perturb the CRD to initiate opening of the autoinhibited state. Rather, KRAS-driven recruitment of RAF to the membrane promotes extraction of the CRD from its autoinhibitory site to initiate opening and activation.

As a next step in our effort to structurally dissect this process, we have determined the structure of KRAS bound to the full-length BRAF/MEK/14-3-3 complex using cryo-EM. The structures reveal a preactivation intermediate, in which key autoinhibitory interactions of the CRD domain with the 14-3-3 dimer and BRAF kinase are unperturbed, despite the engagement of KRAS. Biochemical reconstitutions show that binding of farnesylated KRAS in a liposome environment induces BRAF activation, while the isolated KRAS GTPase domain does not.

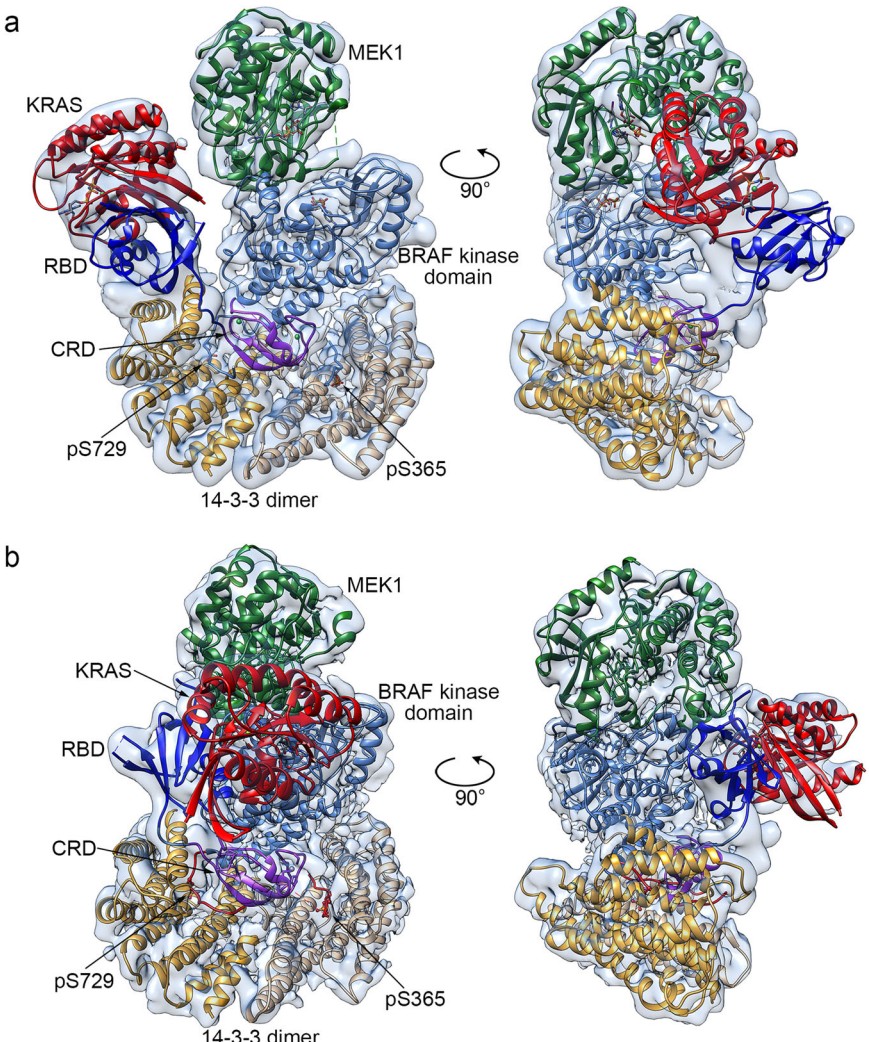

**Fig. 2 | Cryo-EM structures of a KRAS/BRAF/MEK1/14-3-3 complex. a** Structure of the KRAS/BRAF/MEK1/14-3-3 complex in the "RAS up" conformation. A ribbon diagram of the structure, colored as in Figure 1a, is shown together with the transparent cryo-EM density map. **b** Structure of the KRAS/BRAF/MEK1/14-3-3 complex in the "RAS front" conformation, determined with the BS3 cross-linked sample. A ribbon diagram of the structure, colored as in (**a**) above, is shown together with the transparent cryo-EM density map.

## Results

### Cryo-EM structures of a KRAS/BRAF/MEK1/14-3-3 complex

Addition of the KRAS GTPase domain (KRAS$^{GTPase}$, residues 1–169, stabilized in its active state by loading with the non-hydrolyzable GTP analog GMP-PNP), to the autoinhibited BRAF/MEK/14-3-3 complex resulted in formation of a stable complex, as judged by co-elution on size-exclusion chromatography (Fig. 1b). The elution volume was little changed as compared with that of the autoinhibited BRAF complex alone, suggesting that addition of KRAS had not triggered conversion to the active dimeric state. A single-particle cryo-EM reconstruction at ~4.3 Å resolution revealed that the complex indeed retains an auto-inhibited configuration, with KRAS bound to the BRAF RBD domain and extending "up" from the 14-3-3 dimer, positioning KRAS alongside the C-lobe of the MEK1 kinase domain (Fig. 2a and Supplementary Table 1). The density for both KRAS and the BRAF RBD was somewhat weaker and lower resolution than the remainder of the complex (Supplementary Fig. 1a), indicative of conformational variability of these domains, but we were nevertheless able to confidently position both domains (Fig. 2a and Supplementary Video 1). The interaction between KRAS and the BRAF RBD in this structure is essentially the same as that previously seen in crystal structures of CRAF bound to RAS[15,17]; KRAS uses primarily its "switch I" region to bind the RBD,

forming an anti-parallel β-strand interaction with the β2-strand of the RBD domain (Supplementary Fig. 2a, b). KRAS makes only a glancing contact with the MEK; both its N- and C- termini are positioned within 6–7 Å of the MEK C-lobe, near residues Gly237 to Tyr240 in the MEK1 C-lobe (Supplementary Fig. 2c).

We also imaged a similar sample that was subjected to cross-linking with Bis(sulfosuccinimidyl)suberate (BS3) prior to size-exclusion chromatography (Fig. 1c). Using three-dimensional classification, we obtained two reconstructions with this sample, one in a "RAS-up" conformation as described above, and the other in a "RAS-front" conformation in which the KRAS/RBD module pivots to position KRAS in front of the BRAF and MEK1 kinase domains (Fig. 2b, Supplementary Table 1, Supplementary Fig. 1b, and Supplementary Video 2). The RAS-front conformation is not induced by cross-linking, as we also observed this conformation with the three-dimensional classification of particles in the non-cross-linked sample (Supplementary Fig. 1a). Overall, the resolution of the cross-linked RAS-front reconstruction is ~3.9 Å, but as in the non-cross-linked structure, the resolution in the region of the KRAS/RBD is considerably lower. In this conformation, KRAS binds the RBD in the general manner expected, but there is approximately an 11° difference in their relative orientation (Supplementary Fig. 2d). Both the RBD and KRAS contact the interface between the C-lobes of the BRAF

and MEK kinase domains (Fig. 2b and Supplementary Fig. 2e), but given the limited resolution in this region it is not possible to define specific interactions. Indeed, considering the apparent flexibility of the KRAS/RBD unit in both structures, we do not ascribe a particular significance to either the RAS-up or RAS-front conformations. Rather, they appear to be two orientations of the mobile KRAS/RBD module that are sufficiently populated to allow 3D reconstruction. The distinct positions of the RBD in these two structures and in the recent KRAS-free autoinhibited structure[7] are illustrated in Supplementary Fig. 3. Attempts to further define the flexibility of the KRAS/RBD region with 3D variability analysis were not informative; density for this region fades between the two conformations (Supplementary Video 3).

Comparison of the present RAS-up and RAS-front structures with our previously determined RAS-free autoinhibited structure reveals that the constellation of autoinhibitory interactions among the CRD domain, C-lobe of the kinase domain, and the 14-3-3 dimer and its cognate phospho-serine recognition sites is unchanged. Minor differences in the CRD domain in the present structures do not reflect differences due to binding of KRAS. Rather they are a result of an improved fit to the cryo-EM maps enabled by reference to an Alphafold2 model[22] of this domain and modestly improved resolution in this region for the RAS-bound structures.

We further probed inter-domain interactions in this complex with mass spectrometry analysis of the BS3-cross-linked sample (Supplementary Fig. 4 and Source Data). We identified two crosslinks just N-terminal to the RBD domain that are consistent with the RAS-front conformation. These crosslinks connect Lys150 in BRAF to MEK residues Lys185 and Lys353. We identified one additional crosslink within the BRAF RBD domain proper (connecting Lys 183 in the RBD to Lys 253 in the CRD domain), and only one inter-domain crosslink to KRAS (connecting Lys147 in KRAS with Lys353 in MEK1). These latter two crosslinks are inconsistent with both the RAS-up and RAS-front conformations in our cryo-EM reconstructions (Supplementary Fig. 4) and are therefore suggestive of flexibility or additional positions of the KRAS/RBD region.

## Dissecting the role of the RBD in RAF autoinhibition and activation

Recent crystal structures of the CRAF RBD/CRD region in complex with KRAS and HRAS show interactions of the CRD domain with RAS in addition to the well-characterized primary interaction with the RBD[17,18]. These interactions of RAS with the RBD and CRD are mutually exclusive with the interactions of the CRD domain in the autoinhibited state, suggesting the possibility that binding of KRAS to the autoinhibited state could promote its "opening"—release of some or all of the interactions of the CRD with the kinase domain and 14-3-3 dimer[7,17], which is expected to be a key step in the activation process. While the RBD was poorly ordered in our prior cryo-EM structure of autoinhibited BRAF[6], leading us to propose that it was fully exposed for binding by KRAS, recent structures of autoinhibited BRAF from Martinez Fiesco et al. model the RBD in a defined orientation in which binding of KRAS would lead to modest steric clashes with the 14-3-3 domain[7]. This observation, in light of the crystal structures noted above, led to the suggestion that upon recruitment of BRAF to the membrane by KRAS, KRAS engagement could promote release of autoinhibitory interactions by sterically disrupting the RBD/14-3-3 interactions, and perhaps in turn those of the adjacent CRD[7].

One difference in these studies of autoinhibited BRAF is the expression system; we prepared our complex in insect cells, while that studied by Martinez Fiesco et al. was prepared in a mammalian expression system. Although the latter yielded a complex with mammalian rather than insect cell 14-3-3 isoforms, this difference is not expected to affect the RBD interaction because the eight 14-3-3 residues in this interface are near-identically conserved across both insect cell isoforms, and all 7 human 14-3-3 isoforms (only one conservative

substitution across eight positions in nine sequences). While the 14-3-3 side of the interface is conserved, the RAF RBD side is not; four of five RBD residues in the interface diverge across human ARAF, BRAF, and CRAF[7]. Differences in posttranslational modification could also lead to differences in RBD conformation, whether due to expression system, cell culture conditions or purification strategy.

To better understand the contribution of the RBD to the stabilization of the autoinhibited state, we imaged two N-terminal truncations of BRAF, both as BRAF/MEK1/14-3-3 complexes prepared in a manner analogous to our full-length autoinhibited complex[6]. The first, Δ155-BRAF (residues 155–766), removes the unstructured N-terminal region of BRAF including serine 151, a known phosphorylation site near the N-terminus of the RBD that has been proposed to contribute to negative-feedback regulation of RAF[23]. We find that our autoinhibited BRAF complexes are highly phosphorylated on this site, leading to the question of whether it might affect interactions or order of the RBD domain. Single-particle cryo-EM reconstruction of the Δ155-BRAF/MEK1/14-3-3 complex reveals an overall conformation essentially the same as that we previously observed for full-length BRAF complexes (Supplementary Fig. 5a). In both this map and in our previously reported cryo-EM map with full-length BRAF, we observe additional weaker density in the region of the RBD that is insufficiently resolved to allow modeling of this domain (compare Supplementary Fig. 5a, b). This density roughly corresponds to the position of the RBD as modeled in the Martinez Fiesco structure. These authors do not report the phosphorylation state of Ser151 in the mammalian-expressed BRAF used in their study, but irrespective of its phosphorylation state, this residue is also disordered in their structure and does not appear to contribute to autoinhibitory interactions of the RBD. Thus, available structural evidence does not support a role for the phosphorylation of Ser151 in modulating the autoinhibitory interactions of RBD domain.

The second truncation we studied removes the RBD domain as well (ΔRBD-BRAF, residues 233–766). While we were unable to obtain a 3D reconstruction with this sample, the most abundant 2D class averages clearly show the configuration of the autoinhibited complex (Supplementary Fig. 5c). Thus, BRAF can adopt the autoinhibited state even in the absence of the RBD domain. However, in support of a role for the RBD in contributing to the stability of the autoinhibited state, we do note a subjective increase in particle heterogeneity in this sample, including 2D classes in which the CRD domain appears to have been released from its 14-3-3 interactions (Supplementary Fig. 5c).

To probe the accessibility of the RBD for KRAS binding, we measured the affinity of KRAS for various fragments of BRAF and for full-length BRAF in both autoinhibited and active states using microscale thermophoresis (MST, Table 1, Supplementary Fig. 6). The affinity of GMP-PNP loaded KRAS for full-length BRAF in the active, dimeric state (~106 nM) was similar to that in the autoinhibited state (~127 nM), and essentially identical to its affinity for an isolated N-terminal fragment containing both the RBD and CRD domains (~108 nM). For the isolated RBD domain, we measured an affinity of ~26 nM. These measurements point to the RBD domain as the primary contributor to the affinity of BRAF for KRAS. The standard deviations of these MST measurements are somewhat large, perhaps due to limitations in the highest RAF protein concentrations achievable. Nevertheless, the affinity values we obtain agree well with previous studies of KRAS binding to various BRAF and CRAF constructs (Table 1), including measurements of binding to the autoinhibited BRAF complex[7,17,24]. Furthermore, the similar affinity of KRAS for the autoinhibited and active states of full-length BRAF indicate that there is little if any steric hindrance to the engagement of the RBD in the autoinhibited state.

## In vitro reconstitution of BRAF activation

KRAS and other RAS isoforms are modified at their C-terminus by sequential attachment of a farnesyl or geranylgeranyl group, cleavage of the three C-terminal residues, and carboxymethylation[25]. Prenylation

**Table 1 | MST measurement of binding affinity of KRAS to BRAF fragments and complexes**

| BRAF construct/complex (residues) | $K_d$ (nM) | Literature values (nM) | |
|---|---|---|---|
| | | **BRAF** | **CRAF\*** |
| BRAF RBD (151–232) | 26.4 ± 13.9 | 11.2 (ref. 24) | 21.2 (ref. 24) <br> 356 (ref. 17) |
| BRAF BRS-RBD-CRD (39–320) | 107.9 ± 60.1 | 64.6 (ref. 24) | 190.5 (ref. 24) <br> 152 (ref. 17) |
| BRAF BRS-RBD-CRD-C2 (39–434) | 180.6 ± 123.6 | | 188 (ref. 24) |
| BRAF CRD-kinase (233–766) with 14-3-3 | ND | | |
| BRAF/14-3-3 (full length, active dimer) | 105.5 ± 58.7 | 80.2 (ref. 24)† | |
| BRAF/MEK/14-3-3 (autoinhibited) | 127.3 ± 77.2 | 85/127 (ref. 7)‡ | |

\*CRAF lacks a BRS domain; these studies measured binding to RBD or RBD-CRD constructs with SPR.

†The activation state and 14-3-3 binding of the full-length BRAF in this study was not characterized.

‡Affinities were measured for binding of KRAS to autoinhibited BRAF complexes with and without MEK, respectively, using fluorescence polarization.

Source data are provided in the Source Data File.

promotes membrane association of RAS and has long been known to be a requirement for its activation of RAF[20,21]. Furthermore, the addition of the prenylation-promoting CaaX motif to RAF itself is sufficient to drive its membrane localization and activation, independent of RAS[20,21]. Thus it is not surprising that the RAF/MEK/14-3-3 complex is maintained in its autoinhibited configuration when bound to KRAS[GTPase], as this KRAS construct lacks the C-terminal hypervariable region and prenylation site.

To further explore the activation of RAF by KRAS and interactions with lipid in vitro, we developed a reconstitution assay in which activation of the autoinhibited BRAF/MEK/14-3-3 complex is detected by measurement of ERK2 phosphorylation. This assay employs autoinhibited complexes prepared with wild-type MEK1 (MEK1[WT]), rather than the MEK1[SASA] mutant used for structural analysis, and thus does not require exogenous MEK1. In addition to GMP-PNP KRAS[GTPase], we prepared intact KRAS in its fully modified, farnesylated state (KRAS4b-FME)[26], and assessed its ability to activate the autoinhibited BRAF complex in the presence or absence of phosphatidylserine-containing liposomes (PS-liposomes). Phosphatidylserine has previously been shown to be an important determinant of the interaction of RAF with membranes via the CRD domain[27,28]. As expected, we did not observe detectable activation by GMP-PNP KRAS[GTPase] in this assay (Fig. 3a). The addition of KRAS4b-FME induced only a very modest and somewhat variable degree of activation (Fig. 3a, b). The addition of PS-liposomes to the autoinhibited RAF/MEK/14-3-3 complex also resulted in a modest increase in ERK2 phosphorylation, but the combination of PS-liposomes and KRAS4b-FME yielded much more robust activation (Fig. 3b). We note however that the level of activity observed with liposome-associated KRAS4b-FME was markedly less than that of purified active, dimeric BRAF (Fig. 3c), suggestive of incomplete activation of BRAF under these conditions.

## Discussion

RAF activation is a complex, multistep process. Over the past few years, structural studies have provided snapshots of distinct states in this process, including the fully autoinhibited and active states, and have also illuminated the details of interactions of RAS with fragments of RAF at the membrane. Considering that the KRAS-engaged BRAF complex maintains its autoinhibited configuration, the structures described here are best considered as "recruitment complexes" or views of a pre-activation intermediate. Collectively, these structures afford an increasingly detailed molecular understanding of the activation process (Fig. 4).

An important outstanding question is how "opening" of the autoinhibited complex is promoted in the membrane environment. It is clear from the present work that steric effects resulting from binding to KRAS are not sufficient to open the autoinhibited state, although they can contribute by disrupting the stabilizing interactions of the RBD with the 14-3-3 dimer seen in a prior structure[7]. We can conceive of two general mechanisms that could contribute to opening. In one, physical interactions of the recruitment complex with the membrane could directly induce its opening, whether by steric, interfacial effects, or via specific interactions. In the second, proximity to the membrane could promote the association of the CRD with KRAS and the membrane upon "breathing" (transient opening or partial opening) of the complex, thereby increasing the fraction of time in the open state.

Protein factors at the membrane are also important. In particular, the SHOC2 phosphatase complex contributes to RAF activation by dephosphorylating the autoinhibitory 14-3-3 binding site (pS365 in BRAF, pS259 in CRAF)[29,30]. Recent structural and biochemical studies with SHOC2 show that the ternary complex of SHOC2, MRAS, and the PP1c phosphatase assembles in a highly cooperative manner to form a compact structure with mutual interactions among all three components[31–33]. The assembled complex can efficiently dephosphorylate phosphopeptides based on the BRAF pS365 site, as well as the pS365 site in full-length BRAF/14-3-3 complexes in the active, dimeric state[31–34], where this site is exposed. However, it cannot efficiently dephosphorylate the pS365 site in the autoinhibited BRAF/MEK1/14-3-3 complex[31], because the site is buried by interactions with the CRD domain and 14-3-3 dimer. How then is this site accessed, and why is SHOC2 phosphatase activity important for RAF activation if RAF must already be open for the phosphatase to act? Perhaps the simplest model consistent with available data is that in a membrane-associated context, there exists a dynamic conformational equilibrium between the closed state, as observed in the KRAS-bound recruitment complex we describe here, and a hypothesized "open state" in which autoinhibitory interactions are released and pS365 is exposed. Although autoinhibitory interactions are released in the open state, the RAF kinase domain remains monomeric and therefore inactive. SHOC2-mediated dephosphorylation of pS365 in the open state would foreclose the possibility of return to the autoinhibited state, and thereby promote accumulation of open monomers, which would in turn convert to active dimers via 14-3-3-driven dimerization (Fig. 4). This proposed open state of RAS-bound BRAF has not been directly observed, and additional studies will be required to vet this model.

While we do observe activation of the autoinhibited BRAF complex in our in vitro reconstitutions with KRAS-FME in a membrane environment, it appears to be incomplete, as judged by comparison with activity of purified BRAF/14-3-3 dimers[6]. An obvious next direction is incorporation of the SHOC2 phosphatase complex and other factors in our reconstitution system, which will allow us to further dissect the activation process. In addition to SHOC2, kinase suppressor of RAS proteins (KSR1/2) may also contribute to BRAF activation via heterodimerization and interactions with the N-terminal BRAF-specific domain[1,35].

The structures described here show that engagement of BRAF by KRAS affects the position and interactions of the RBD domain, but does

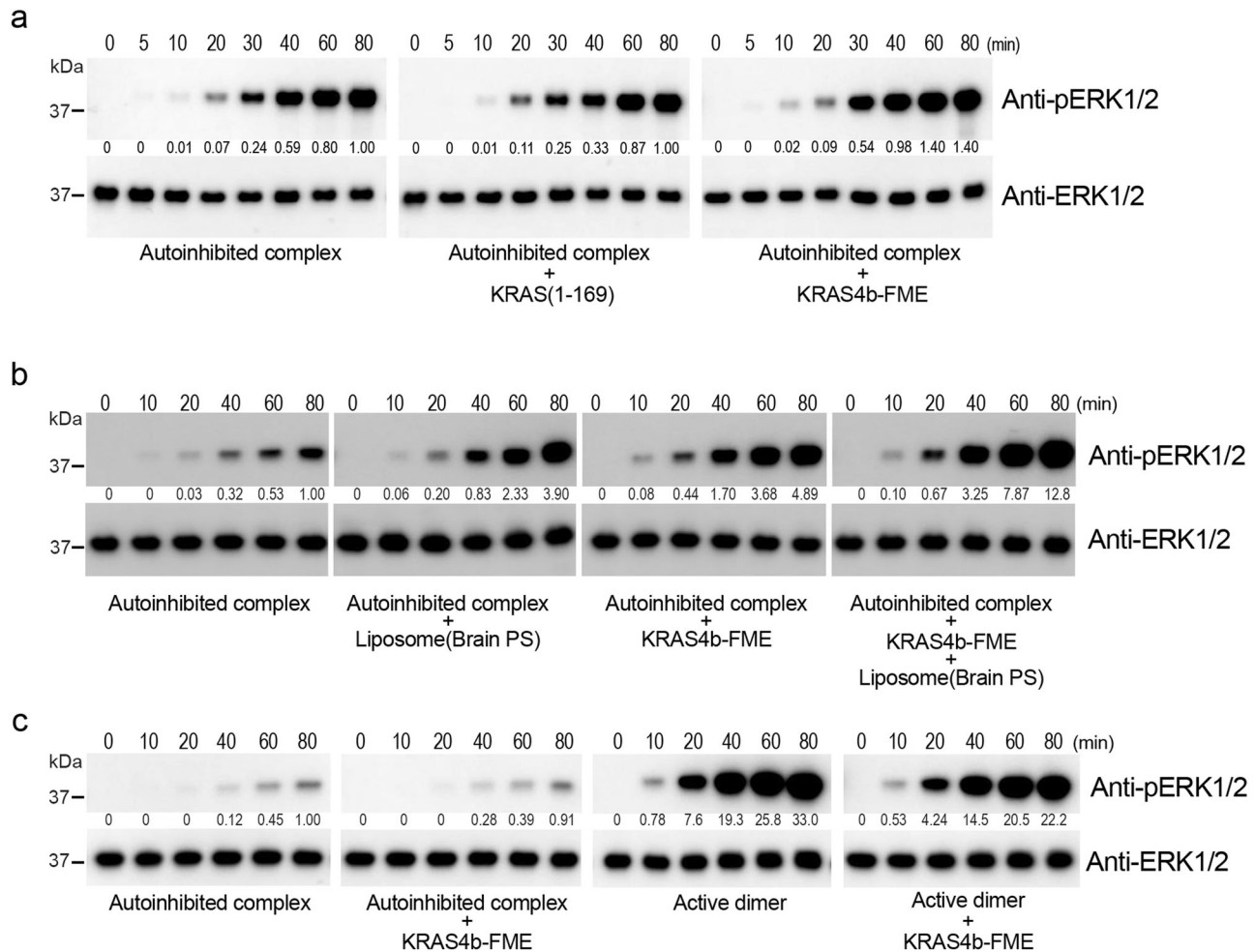

**Fig. 3 | Activation studies of the autoinhibited BRAF/MEK1/14-3-3 complex.**
**a**–**c** The time course of phosphorylation of ERK2 by the purified BRAF/MEK1/14-3-3 complex, with or without the addition of the indicated KRAS construct and/or liposomes, is measured by western blotting with an anti-pERK1/2 antibody. Integrated band intensities, normalized to the intensity of the autoinhibited complex alone at the 80 min time point, are shown under the blots. **a** Activity of the autoinhibited complex alone, or with the addition of either GMP-PNP loaded KRAS^GTPase or KRAS4b-FME at a 1:1 molar ratio. **b** Activity of the autoinhibited complex alone or with the addition of phosphatidylserine (PS) liposomes, KRAS4b-FME, or both liposomes and KRAS4b-FME, as indicated. Liposomes were added to a final

concentration of 0.1 mg/ml, and GMP-PNP loaded KRAS4b-FME was added in a 2:1 molar ratio to the autoinhibited BRAF/MEK1/14-3-3 complex (5 nM). **c** Comparison of the relative activity of the BRAF/MEK1/14-3-3 complex in the autoinhibited vs. active, dimeric state, with or without the addition of KRAS4b-FME. The samples for the autoinhibited complex with and without KRAS4b-FME are aliquots of the same reaction as the corresponding samples in (**b**), re-run on the same gel and blot with the active dimer reactions to allow comparison of their relative activities. Note that the exposure of the blot in (**c**) (5 s) is much shorter than that in (**b**) (2 min). Original uncropped blots are provided in the Source Data File. Experiments in (**a**–**c**) were performed three times each with similar results.

not directly disrupt core autoinhibitory interactions among the CRD domain, BRAF kinase domain, and the 14-3-3 and its recognition sites. Thus, binding and activation by KRAS are distinct steps. Considerable effort has been devoted to blocking engagement of RAF and other effectors by RAS, with little success apart from recently developed covalent KRAS G12C inhibitors[36]. This is not surprising, as high-affinity protein-protein interactions are notoriously difficult to disrupt with small molecule inhibitors. The separability of KRAS binding and RAF activation illustrated here suggests stabilization of the recruitment complex as a potentially attractive alternative approach. A small molecule with a "splinting" or "molecular glue" mechanism[37] that stabilized interactions of the 14-3-3 domain with the kinase C-lobe or CRD domain could block RAF activation by preventing opening of the autoinhibited state, which is a prerequisite for pS365 dephosphorylation and RAF dimerization and activation.

## Methods
### Preparation of KRAS proteins
Recombinant KRAS^GTPase (residue 1–169) was prepared as described previously[17]. For the production of full-length farnesylated and

methylated KRAS4b (KRAS-FME), a customized baculoviral expression system was obtained from Dr. D. Esposito (NCI-Frederick), allowing isolation of the fully processed KRAS4b protein from infected Sf9 insect cells (Gibco Sf9 cells, cat.no. 11496015) as described previously[26].

### Preparation of KRAS^GTPase/BRAF^WT/MEK1^SASA/14-3-3 complex
The BRAF^WT/MEK1^SASA/14-3-3 complex was prepared in an autoinhibited state as described[6]. GMP-PNP-loaded KRAS^GTPase was added to the purified BRAF^WT/MEK1^SASA/14-3-3 complex in a 1.2-fold molar excess. The mixture was incubated for 30 min on ice, then applied to a size-exclusion chromatography column (Superose 6 increase 10/300, Cytiva) to remove excess KRAS protein. Size-exclusion chromatography was performed in SEC buffer (50 mM Tris pH 7.5, 150 mM NaCl, 2 mM MgCl_2, 0.5 mM TCEP, 10 μM ATP-γS, 2 μM GDC-0623, 10 μM GMP-PNP).

### Preparation of cross-linked KRAS^GTPase/BRAF^WT/MEK1^SASA/14-3-3 complex
KRAS^GTPase was added to the BRAF autoinhibited complex as described above, incubated on ice for 30 min, then loaded onto size-exclusion chromatography in cross-linking buffer (20 mM HEPES pH 7.5, 150 mM

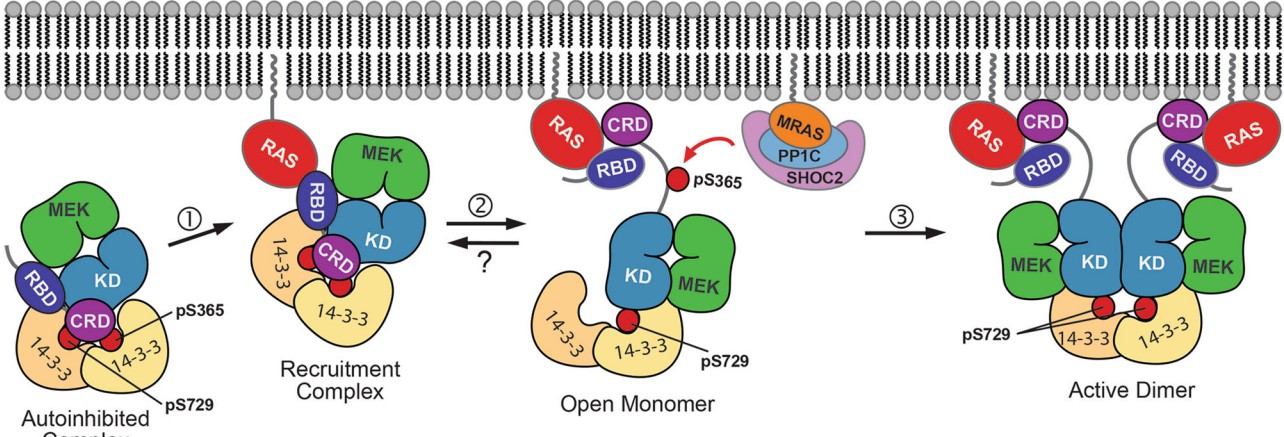

**Fig. 4 | Assembling a model for RAS-driven activation of RAF.** In the quiescent state, BRAF is maintained in an autoinhibited complex with MEK and a 14-3-3 dimer in the cytosol. As a first step in activation (1), this complex is recruited to the membrane by GTP-bound RAS to form a recruitment complex, as visualized in this study. The interaction with farnesylated RAS in a membrane context induces release of autoinhibitory interactions in the BRAF/MEK1/14-3-3 complex, resulting in formation of an "open" monomer complex (2). This second step in activation may result from extraction of the CRD domain from its relatively buried site in the autoinhibited complex, due to preferential binding to RAS and the plasma membrane. Opening exposes pS365 for dephosphorylation by the SHOC2 phosphatase complex, and also allows the 14-3-3 dimer to rearrange to bind the C-terminal pS729 site of two BRAF molecules, driving and stabilizing the active BRAF dimer (3). Because the pS365 site is buried and not accessible for dephosphorylation in the recruitment complex, and because S365 plays no known role in the formation of the active dimer, we hypothesize that the SHOC2 phosphatase complex acts on the open monomer state and contributes to activation by preventing its "reclosure". Once pS365 is dephosphorylated, BRAF cannot reassume the closed, autoinhibited state observed in the recruitment complex, and the resulting accumulation of open monomers favors rearrangement into active dimers. Protein Data Bank accession codes for structures supporting distinct states and components of this model include: 6NYB and 7MFD (autoinhibited complex); 8DGS and 8DGT (recruitment complex, this study); 6PP9 and 6U2G (BRAF/MEK kinase domain portion of open monomer); 6XHB and 7JHP (RAS complexes with RBD/CRD fragments of RAF, relevant to the open monomer and active dimer states); 6Q0J, 6Q0K, 6UAN, 6XAG, 7MFF (14-3-3-bound BRAF dimers, with or without MEK); and 7SD0, 7UPI, 7TXH, and 7TVF (ternary SHOC2, MRAS, PP1C phosphatase complex).

NaCl, 2 mM MgCl₂, 0.5 mM TCEP, 10 µM ATP-γS, 2 µM GDC-0623, 10 µM GMP-PNP). Peak fractions containing the complex were pooled and concentrated to a volume of 500 µl and an approximate concentration of 5 µM for the protein complex. Freshly prepared bis(sulfosuccinimidyl)suberate (BS3) solution was added to the concentrated complex to a final concentration of 1 mM and incubated for 45 min at room temperature, followed by quenching of the cross-linking reaction by the addition of 100 mM Tris-HCl (pH 7.5) for 15 min. Finally, the cross-linked KRAS/BRAF/MEK1/14-3-3 complex was subjected to a "polishing" round of size-exclusion chromatography in SEC buffer to remove unreacted BS3 and any aggregated or oligomerized complex.

**Preparation of autoinhibited complexes with truncated BRAF**
The BRAF$^{\Delta155}$/MEK1$^{WT}$/14-3-3 and BRAF$^{\Delta RBD}$/MEK1$^{SASA}$/14-3-3 complexes were prepared essentially the same as described previously for full-length BRAF in an autoinhibited state[6], except that truncated BRAF constructs were employed. Briefly, recombinant baculovirus expressing the N-terminally truncated BRAF constructs (Δ155-BRAF, residues 155−766 or ΔRBD-BRAF, residues 233−766 of human BRAF) was co-expressed with MEK1$^{WT}$ or MEK1$^{SASA}$, respectively, in baculovirus/insect cell expression system with Sf9 cells by mixing separate baculoviruses. The complex of BRAF, MEK1, and insect cell derived 14-3-3 ε, ζ co-purified through Ni-NTA- and Strep- affinity chromatography and finally through size-exclusion chromatography in 50 mM Tris pH 7.5, 150 mM NaCl, 2 mM MgCl₂, 0.5 mM TCEP, 10 µM ATP-γS, and 2 µM GDC-0623.

**Cryo-EM data acquisition and processing**
KRAS$^{GTPase}$/BRAF$^{WT}$/MEK1$^{SASA}$/14-3-3 complex in SEC buffer was applied to glow-discharged holey carbon grids (Quantifoil R1.2/1.3, 400 mesh) and vitrified using a Leica EM GP. Frozen-hydrated samples were imaged on an FEI Titan Krios at 300 kV with a Gatan Quantum Image Filter with K3 direct detection camera in super-resolution mode with a total exposure dose of ~45 electrons using

SerialEM. 40 frames per movie were collected at a magnification of ×105,000, corresponding to 0.85 Å per pixel. In total, 8148 micrographs were collected at defocus values ranging from −1.8 to −2.8 µm. Two separate data sets were acquired from two grids of the same sample and the dataset with more images (8148) was chosen for single-particle reconstruction. Cross-linked KRAS$^{GTPase}$/BRAF$^{WT}$/ MEK1$^{SASA}$/14-3-3 complex with BS3 was vitrified and imaged as above on the Krios in counting mode. Total dose of 50 electrons and 50 frames per movie were collected at 0.825 Å per pixel with defocus value ranging −1.5 to −2.5 µm from two data collections of 11178 and 11144 images (from two grids prepared from the same sample). The BRAF$^{\Delta155}$/MEK1$^{WT}$/14-3-3 complex was vitrified using a FEI Vitrobot Mark IV and frozen-hydrated samples were imaged on the Krios with phase plate in super-resolution mode at 0.53 Å per pixel with a total exposure dose of ~50 electrons. Total 3252 micrographs collected at defocus values ranging from −0.2 to −1.2 µm using serialEM from one grid. Details of the data collection and dataset parameters are summarized in Supplementary Table 1.

Dose-fractionated images were gain normalized, aligned, dose-weighted and summed using MotionCor2 (cross-linked sample and Δ155-BRAF complex) or the motion correction implementation within Relion (non-cross-linked sample)[38]. Contrast transfer function (CTF) and defocus value estimation were performed using CTFFIND4[39] for the cross-linked sample and Patch CTF estimation within CryoSPARC for the non-cross-linked data[40]. Workflow for the single-particle reconstructions from the non-cross-linked, cross-linked and Δ155-BRAF/MEK1$^{WT}$/14-3-3 samples is shown schematically in Supplementary Figs. 7−9, respectively. In brief, particle picking was carried out using crYOLO[41] followed by initial 2D classification within Relion[42] to give 570,743 and 3,989,095 particles for the non-cross-linked and cross-linked samples, respectively. Multiple rounds of 2D and 3D classification was then carried out on the sample to remove "junk" particles and to identify particles with distinct KRAS positions. This resulted in 69,377 particles in the non-cross-linked samples in the "Up"

conformation which led to a 4.3 Å reconstruction following non-uniform refinement in CryoSPARC[43]. For the cross-linked sample, 190,489 particles were identified with KRAS in the "Front" position resulting in a 3.9 Å reconstruction from non-uniform refinement. A subset of Δ155-BRAF/MEK1[WT]/14-3-3 micrographs were removed through a mixture of CTF resolution filtering and manual inspection to leave 1,513 images. Particle picking was performed in RELION using a 30 Å low-pass-filtered autoinhibited BRAF map (obtained initially from ab initio refinement of a subset of the data picked with Laplacian reference free picking in RELION), the reference-based picking resulted in 536,128 particles. 2D classification in RELION was used to remove junk particles and false positives, leading to 384,752 particles for further analysis in 3D classification. Two rounds of 3D classification in RELION resulted in a homogeneous population of 91,490 particles with clear secondary structure elements visible in the 3D reconstruction. This was then subjected to non-uniform refinement in CryoSPARC, resulting in a 4.1 Å reconstruction.

Statistics for the final refinement are presented in Supplementary Table 1. A representative image and 2D class averages, together with "gold-standard" Fourier shell correlation plots and heat maps showing the distribution of particle orientations for all structures are presented in Supplementary Fig. 10.

Structural biology applications other than CryoSPARC used in this project were compiled and configured by SBGrid[44]. Models were fit into the map using Coot[45] and further refined with PHENIX[46].

### BRAF activity and KRAS/BRAF/MEK/ERK pathway reconstitution assays

BRAF activity assays with autoinhibited complex (BRAF[WT]/MEK[WT]/14-3-3), and active dimer (BRAF[WT]/14-3-3) were performed in assay buffer (20 mM HEPES, pH 7.4, 150 mM NaCl, 10 mM MgCl$_2$, 1 mM TCEP, 1 mM NaVO$_4$) at 25 °C. The final enzyme complex concentration in the reactions was 5 nM. Equimolar wild-type MEK1 protein was added to BRAF[WT]/14-3-3 complexes for further reaction. ERK2 protein at a concentration of 2 µM was used as substrate. KRAS at a concentration of 10 nM of (a 2:1 molar ratio with the BRAF complex) was added to reaction mixture in RAS activation experiments. Reactions were initiated by adding ATP to a final concentration of 1 mM and quenched at the indicated time points by adding 2× SDS sample buffer followed by heat inactivation for 5 min at 85 °C. Assay results were analyzed by western blot with Anti-ERK (Cell signaling technology, #9102, 1:1000 dilution) and pERK (Cell signaling technology, #45899, 1:1000 dilution) antibody. For activation experiments that included liposomes, liposomes were prepared by hydrating freeze-dried Brain Phosphatidylserine (Avanti, #840032) with a buffer containing 20 mM Hepes, pH 7.4, and 150 mM NaCl to a concentration of 2.5 mg/ml. The hydrated lipids were frozen in liquid N$_2$ and thawed at 50 °C a total of 5 times, then passed through a mini-extruder ten times to make large uni-lamellar vesicles. The activation experiments with liposomes were performed in the presence of 0.1 mg/ml liposomes (final concentration), and the enzyme reactions and western blotting were carried out as described above. Representative original uncropped blots are provided in the Source Data file.

### Preparation of KRAS and BRAF proteins for microscale thermophoresis studies

DNA encoding residues 1–169 of KRAS4b was cloned into a modified pET vector for expression with a TEV-cleavable N-terminal His$_6$-tag in *E. coli*. The sequence of the TEV cleavage site was modified from ENLYFQS to ENLYFQC so that after cleavage to remove the His$_6$-tag, KRAS is left with an N-terminal cysteine residue for purposes of fluorescent labeling (see below). The BRAF RBD domain (residues 151–232) was also cloned into the modified pET vector for expression with a TEV-cleavable N-terminal His$_6$-tag in *E. coli*. The resulting plasmids were transformed into BL21(DE3) cells. Protein expressions were

induced by the addition of 1 mM isopropyl-β-ᴅ-thiogalactoside at 20 °C for 18 h. Cells were harvested by centrifugation and resuspended in 20 mM HEPES (pH 7.0), 200 mM NaCl, 10 mM imidazole. After sonification, the cell lysate was applied to Ni-NTA agarose beads (Qiagen) and then eluted with 300 mM imidazole. Longer BRAF constructs that included the CRD region (residues 39–320, 39–434, and 233–766) were cloned into the modified baculovirus transfer vector pAc8 with a TEV-cleavable N-terminal His$_6$-tag. These CRD-containing BRAF proteins were expressed by baculoviral infection of Sf9 insect cells and purified by Ni-NTA affinity essentially as described above. Full-length BRAF/14-3-3 and BRAF/MEK1/14-3-3 complexes in active and auto-inhibited states, respectively, were prepared as previously described[6].

### Fluorescent labeling and GMP-PNP loading of KRAS for microscale thermophoresis studies

KRAS was labeled with Alexa Fluor 647 at its amino terminus following the protocol described in ref. 47. Briefly, to expose the N-terminal cysteine residue, the His$_6$ tag was removed by TEV protease. During the cleavage procedure, a "one-pot" reaction with MESNA (Sodium 2-mercaptoenthanesulfonate, Sigma Aldrich M1511) and Alexa Fluor™ 647 NHS Ester (Invitrogen cat# A20006) was initiated. This procedure yielded KRAS selectively labeled with Alex Fluor 647 via an amide bond with the N-terminal cysteine residue. After the labeling reaction, the labeled KRAS protein was purified by size-exclusion chromatography using 16/60 Superdex75 increase column (GE Healthcare) pre-equilibrated with 20 mM HEPES (pH 7.0), 150 mM NaCl, 1 mM TCEP.

As purified, KRAS is bound primarily to GDP. In order to load with the non-hydrolyzable GTP analog GMP-PNP for binding studies, we carried out an exchange procedure as described previously[48]. Briefly, purified KRAS protein was mixed with GMP-PNP (molar ratio of 10:1 GMP-PNP:KRAS) and calf intestinal alkaline phosphatase (NEB cat# M0290, 3 units per mg of KRAS). The reaction mixture was incubated for 3 h at room temperature and then purified by size-exclusion chromatography on a 10/300 Superdex75 GL column (GE Healthcare) pre-equilibrated with 100 mM HEPES (pH 7.0), 150 mM NaCl, 1 mM Tris(2-carboxyethyl)phosphine hydrochloride (TCEP). GMP-PNP loaded KRAS was then concentrated, flash frozen, and stored at −80 °C until used.

### Microscale thermophoresis measurements

Binding affinities of the Alexa Fluor™ 647-labeled GMP-PNP-bound KRAS and BRAF proteins were measured using microscale thermophoresis (MST). Before the MST experiment, Tween-20 detergent was added to all samples to 0.05%. All protein samples were stable in the presence of this concentration of Tween-20. In all, 10 µL of serially diluted BRAF proteins from 10 pM to 1 µM were loaded into eight PCR tubes. Then, 10 µL of 5 nM N-terminal-labeled KRAS mixed into each reaction tube. The binding affinity measurements were carried out using the modified manufacturer's protocol (Monolith NT.115pico, NanoTemper Technologies) in the Center for Macromolecular Interactions in Harvard Medical School (Boston, MA). Experiments were performed three times, with independent dilution series, and results were averaged to obtain the reported affinities and error estimates. The data are provided in Source Data File.

### Cross-linking mass spectrometry studies

Complexes cross-linked with BS3 were denatured in 8 M urea and subjected to reduction and carbamidomethylation with TCEP and iodoacetamide. Samples were diluted to 2 M urea and digested with trypsin (1:50 ratio of trypsin:protein) overnight at 32 °C. Digestions were brought to 1% formic acid (FA) and peptides were dried by vacuum centrifugation and desalted over a C18 column. Digests were resuspended in 5% acetonitrile (MeCN)/1% FA and analyzed on an Ultimate 3000 RSLCnano system coupled to an Orbitrap Eclipse mass spectrometer (Thermo Scientific). Peptides were separated across a 55-min linear gradient of 7–22% MeCN in 1% FA, followed by 15 min to 45%

MeCN over a 50-cm C18 column (ES803A, Thermo Scientific, Waltham, MA, USA) and electrosprayed (2.15 kV, 300 °C) into the mass spectrometer with an EasySpray ion source (Thermo Scientific). Precursor ion scans (300–2000 $m/z$) were obtained in the orbitrap at 120,000 resolution in profile (RF lens % = 30, Max IT = 100 ms, 1 microscan). Fragment ion scans were acquired in the orbitrap at 30,000 resolution (1.6 $m/z$ isolation window, HCD at 30% NCE, 5e$^4$ AGC).

Raw data were searched against KRAS, BRAF, and MEK1 construct sequences, as well as endogenous Sf9 14-3-3ε and 14-3-3ζ sequences for BS3-cross-linked peptides using Protein Prospector Batch-Tag (https://prospector.ucsf.edu/prospector/mshome.htm)[49–51] permitting mass accuracy of ±5 ppm for precursors and ±10 ppm for fragment ions, two missed cleavages by trypsin, oxidation of Met, carbamidomethylation of Cys, phosphorylation of Ser/Thr/Tyr, and BS3 mono-linked Lys. By manually inspecting a subset of these search results, we found that implementing the preliminary filter of a score difference >15 narrowed the list to spectra that were more likely to be correct hits, which was a manageable size for manual validation. Representative spectra from each cross-linked species in the resulting list were manually inspected. Any cross-linked spectra that did not contain multiple abundant fragment ions from each linked peptide or did contain multiple unexplained peaks were removed from the list. The resulting peptide spectral match (PSM) file and representative spectra of manually validated inter-and intra-molecular crosslinks are presented in Supplementary Data Files 1, 2. For comparison to an approach that allows for FDR filtering, the data were searched with pLink2 (http://pfind.org/software/pLink/)[52] against a target-decoy database containing complex member sequences and pLink2 default contaminants with the following search parameters: mass accuracy of ±5 ppm for precursors and ±10 ppm for fragment ions, two missed cleavages by trypsin, oxidation of Met, carbamidomethylation of Cys, phosphorylation of Ser/Thr/Tyr. Search results were filtered to 1% FDR at the PSM level. All manually validated crosslinks reported from the Protein Prospector search were also identified in this pLink2 search.

The mass spectrometry proteomics data have been deposited to the ProteomeXchange Consortium via the PRIDE[53] partner repository with the dataset identifier PXD042584.

### Reporting summary

Further information on research design is available in the Nature Portfolio Reporting Summary linked to this article.

## Data availability

Cryo-EM density maps for the structures described here have been deposited in the Electron Microscopy Data Bank (EMDB) and are available with accession codes: EMD-27428 (KRAS-up structure), EMD-27429 (KRAS-front structure), EMD-40253 (Δ155-BRAF/MEK1/14-3-3 structure). Atomic coordinates for the KRAS-bound structures have been deposited in the Protein Data Bank and are available at www.rcsb.org with accession codes: 8DGS (KRAS-up structure), 8DGT (KRAS-front structure). Protein Data Bank accession codes for previously published structures used in this study are 6NYB (Structure of autoinhibited BRAF complex), 6VJJ (Structure of KRAS with CRAF RBD), 7MFD (Structure of autoinhibited BRAF complex). The raw data files, fasta files, peak lists, and search results for our ccrosslinking mass spectrometry studies have been deposited in the PRIDE repository: PXD042584 (cross-linking mass spectrometry data for KRAS/BRAF/MEK1/14-3-3 complex). Source data are provided with this paper.

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

## Acknowledgements

We thank the staff of the Harvard Cryo-EM Center for Structural Biology for assistance with data collection. This work was supported by National Institutes of Health grants R35CA242461 (M.J.E.), PO1CA154303 (M.J.E.), P50CA165962 (M.J.E.) and R50CA221830 (E.P.), and by the PLGA fund of the Pediatric Brain Tumor Foundation. We thank Dr. K. Arnette and Center for Macromolecular Interactions at Harvard Medical School for help with the analysis of MST experiments.

## Author contributions

E.P. expressed and purified the KRAS and BRAF/MEK/14-3-3 complexes and designed and executed the cross-linking and KRAS/BRAF/MEK/ERK pathway activation studies described here, with assistance from S.O. S.R. carried out the single-particle reconstructions. H.J. prepared cryo-EM samples and collected EM data; K.S. collected cryo-EM data. A.S. executed and interpreted the cross-linking mass spectrometry study. B.-W.K. designed and executed the microscale thermophoresis binding studies. E.P. and M.J.E. built the atomic models. H.J. and M.J.E. directed the project, and M.J.E. drafted the manuscript with input from all authors.

## Competing interests

M.J.E. receives or has received sponsored research support from Novartis, Sanofi, Takeda, and Springworks Therapeutics and consulting income or honoraria from Novartis, H3 Biomedicine and Ikena Oncology. The remaining authors declare no competing interests.
