## [Peer Review File · Nature Communications]

REVIEWER COMMENTS

Reviewer #1 (Remarks to the Author):

The manuscript entitled 'Cryo-EM structure of a RAS/RAF recruitment complex' by Park et al. describes cryo-EM reconstructions of KRAS bound to intact BRAF in an autoinhibited state with MEK1 and a 14-3-3 dimer. The authors solve reconstructions of the complex and find the Ras-binding domain (RBD) of BRAF in two different orientations. Given the reduced density/lower resolution for both KRAS and BRAF RBD relative to the rest of the complex, the authors postulate conformational variability of the RBD. These findings are in contrast to a recent Cryo-EM structure solved by Martinez Fiesco et al (2022) in which the RBD forms contacts with 14-3-3 in an apparently more fixed conformation, suggesting that RAS interactions with 14-3-3 may obscure binding to the RBD.

Herein, the investigators postulate that BRAF autoinhibitory contacts are unperturbed by RAS binding to the RBD given the conformational dynamic nature of the RBD. To obtain further support for their hypothesis, the authors conduct a series of in vitro binding measurements and activation assays. Consistent with their Cryo-EM data, they find similar affinity of RAS for RAF between the autoinhibited and active BRAF complexes with significant BRAF activity observed only in the context of processed KRAS in the presence of liposomes. Hence, they propose that when RAS engages the RBD, the CRD autoinhibitory contacts with 14-3-3 and the kinase domain remain intact (intermediate state) and that RAS binding alone is not sufficient to activate the CRAF.

The cryo-EM reconstructions of KRAS bound to intact autoinhibited BRAF described in this manuscript, lend new insights into what the investigators term a BRAF 'pre-activation intermediate'. Moreover, in conjunction with RAS binding and RAF activation assays, their findings support previous work in the field that RAS binding to the RAF-RBD localizes cytosolic RAF to the membrane but additional steps are required to promote activation. However, some concerns exist with the manuscript in its present state. As some of the conclusions in current study are based on indirect readouts and the Cryo-EM reconstruction solved in the absence of the membrane, the manuscript would be strengthened by summarizing/referencing the previous body of work that RAS binding alone is not sufficient to activate RAF to better lay the foundation for their hypothesized intermediate. Moreover, error analysis is missing for the binding measurements and the activity measurements lack quantification. The manuscript would also be strengthened by the inclusion of more detailed information regarding RAS/RBD and CRD autoinhibitory interactions in the reconstructions. Mutational analyses (binding/activation assays) of the previously proposed autoinhibitory RAF RBD/14-3-3 contacts may add additional support for the proposed intermediate state and mechanistic interpretation of the results.

More detailed questions/concerns/suggestions are listed below.

There is a large body of data on RAS mediated activation of RAF. Of note, RAS binding to the RAF RBD has been shown to promote membrane association and is a key step yet insufficient for RAF activation. These previous findings should be concisely summarized to lay the foundation for the proposed pre-activation intermediate.

Is the RAS-RAF RBD binding interface retained in the 'preactivation intermediate' (e.g. similar to that determined from previous structural studies)?

Additional information clarifying the nature of the RAS-CRD autoinhibitory contacts and whether these contacts are at all altered upon RAS engagement is needed.

The resolution of KRAS in non-crosslinked and BS3-crosslinked cryo-EM maps is low (8 - 9.5 Å) and limits analysis of the data.

The authors evaluate the binding affinity of RAS with RAF but should compare their findings to previous reported data (some on CRAF). Errors are lacking in the Table associated with Figure 3.

The activation assays are not quantified so that a relative comparison can be made.

Reviewer #2 (Remarks to the Author):

In the manuscript by Park et al., new structural insights about RAS/RAF/MEK/ERK signaling cascade are shown. The importance of this signaling cascade in cancer development is well-documented, but despite all the effort to understand details surrounding this cascade, some important questions are still left unanswered. In this study, authors addressed one of those remaining questions: the effect of RAS binding on the downstream signaling complex composed of RAF, MEK1, and 14-3-3 proteins. In addition, the authors probed whether this is the crucial step in transition of BRAF from an autoinhibited monomeric state to an activated dimeric state. Two cryo-EM structures of RAS/BRAF/MEK/14-3-3 complex, exhibiting the difference in the orientation of RAS protein relative

to the rest of the complex (“RAS-up” and “RAS-front” conformations), are reported. The authors provide the structural evidence that BRAF remains in autoinhibited state upon RAS binding, suggesting that the obtained complex represents a pre-activation intermediate in autoinhibitory state. Interestingly, in their biochemical assays, the authors managed to observe improved BRAF activation upon addition of fully modified KRAS (KRAS4b-FME) and phosphatidylserine-containing liposomes (PS-liposomes). Overall, the manuscript is nicely written and organized and is worthy of publication in Nature Communications. However, before publishing, the authors need to address a few major and minor concerns.

Major concerns:

1. In the reported cryo-EM structures, the authors note conformational variability regarding the RAS positioning. However, the analysis of this structural variability is missing. To address this, the authors should consider performing additional steps in cryo-EM processing (either multi-body refinement or 3D variability analysis) of their data and present the analysis in form of a movie.

2. Along the same lines, on page 5 the authors report inconsistency between reported structures and obtained mass spectrometry data of crosslinks. After completing the suggested 3D variability analysis, it would be interesting to see if the crosslink mass spec data align with the structural flexibility.

3. The authors state on page 4 that the “RAS-front” conformation is not induced by cross-linking, since they can visualize the same conformation in their non-crosslinked sample (Supplementary Fig.1a). However, looking at the provided data processing flowcharts (Supplementary Figure 4 and Figure 5), it is difficult to discern which maps represent the “RAS-front” conformation. Moreover, it is unclear how many particles from the entire particle stack contributed to that conformation/state.

Hence, to aid the reader, “RAS-up” and “RAS-front” maps should be colored differently, the number of particles contributing to both conformations/states should be clearly shown and maps should be highlighted in both non-crosslinked and crosslinked processing schematics (Supplementary Figure 4 and Supplementary Figure 5).

4. In Figure 3, except for the BRS-RBD-CRD plot, it appears that the authors are consistently missing the higher concentration points, suggesting that full saturation of binding sites was not attained under the assay conditions used. This could impact the K_d value estimate and additional titration points should be provided.

5. Equally importantly, have the authors assessed the integrity of each protein in the MST buffer before performing MST experiments? If so, what are the results? If not, then such tests should be performed. This is particularly valuable for binding assays completed under higher concentrations of a binding partner.

Minor concern:

- Figure 1c, right panel – authors should clarify if the lane 3 contains higher amount of the sample loaded, and if so, by how much.

Reviewer #3 (Remarks to the Author):

In this manuscript by Park, Rawson and colleagues, the group reports cryo-EM structures of a KRAS bounding to BRAF/MEK/14-3-3(2) autoinhibited complex purified from baculovirus-infected insect cells. The KRAS/BRAF/MEK1/14-3-3 complex shows KRAS bound to the flexible RAS-binding domain (RBD) of BRAF in two orientations.

In addition, the authors carried out in vitro biochemical activation studies which confirmed that membrane recruitment is needed for BRAF activation, consistent with previous studies. The structures of RAS-BRAF complexes are of general interest to those in the kinase and RAS/RAF signaling field. The results of this study may be suitable for publication in Nature Communications, once the following major concerns have been addressed.

Major comments:

1. The BRAF used in this study is human BRAF expressed and purified from a heterologous (insect) expression system. Differences in post-translational processing in the insect cell expression system (particularly phosphorylation) may differ between the insect-cell expressed protein and the endogenous human protein. This point needs to be discussed in the manuscript. Moreover, can the authors analyze the phosphorylation state of the BRAF protein used in this study and compare it to that expressed from a mammalian expression system? In particular, it is known that the Ser151 of BRAF is maintained in a phosphorylated state due to the high level of basal ERK activation in Sf9 insect cells. As Ser151 is in close vicinity of the RBD, phosphorylation of S151 could directly interfere with the RBD region and Ras binding to BRAF complexes purified from an insect expression system. Have the authors investigated this issue? This point should also be discussed in the manuscript.

2. The system and methodology used in the current study to produce the BRAF autoinhibited complex used to study binding to RAS was previously reported by the same group in Park et al (Nature, 2019), yielding a BRAF autoinhibited structure with an insufficient resolution of the RBD to model the position and orientation of the RBD, indicating it was already released from the autoinhibited complex, flexible, and solvent exposed for RAS binding. In contrast, the RBD domain in the full length BRAF/MEK/14-3-3 and BRAF/14-3-3 autoinhibited complexes from mammalian cells

was previously resolved in Martinez Fiesco et al (Nature Communications, 2022). The authors found that the RBD of BRAF is partially occluded by 14-3-3, requiring structural rearrangements to permit RAS to fully bind to BRAF, which was further validated using mutagenesis experiments. The working system differences as well as the different autoinhibitory complexes should also be discussed in the manuscript.

Martinez Fiesco et al (Nature Communications, 2019) proposed a model for how the monomer to dimer transition in BRAF is mediated by RAS binding in presence of membrane. The monomer to dimer transition model is in the context of membrane presence instead of RAS binding alone and is misstated in several places in this manuscript. Please correct.

3. Similarly, as mentioned before the approximate position of the RBD in the BRAF/MEK1/14-3-3 complex was previously reported in Park et al (Nature, 2019) by the same group, but the resolution was not sufficient to position the known RBD model. Can the authors provide analysis of the RBD region before and after KRAS binding? Are the two orientations of RBD in the KRAS/BRAF/MEK1/14-3-3 complex also observed in the BRAF/MEK1/14-3-3 complex?

4. The novelty of the reported structure is the interaction between RAS and the autoinhibited RAF complex. Unfortunately, the resolution at the RAS-RAF RBD region is very poor (worse than 8Å), lacking well-defined secondary structure, preventing the authors from reporting details of interactions. We strongly recommend the authors to get better quality data for this part of the structure that will allow them to accurately model the RBD domain and RAS protein in their map and draw conclusions.

With new improved maps, can the authors provide semi-transparent map density of the KRAS and RBD regions to help the reader evaluate this interface? Are the KRAS and RBD regions well resolved? What is the local resolution of the critical KRAS and RBD regions? Please include a supplementary figure with details of the cryo-EM density for this interface so the reader can judge if the map allows identification of the KRAS and RBD region.

5. The binding curves for the different BRAF constructs with KRAS seems not to reach saturation what could lead to overestimation of the KD values, we recommend achieving saturation for its determination.

In addition, please give an estimation of the error for the affinities determined using MST. Please include these values in a table. Please provide a supplementary figure with the raw MST traces. Please explicitly state in the methods how many times the experiment was performed. Ideally, MST experiments should be performed three times and the affinities averaged to give a reliable estimation of the affinity value and the error.

6. On page 6, line 189, the authors state “Furthermore, the similar affinity of KRAS for the autoinhibited and active states of full-length BRAF suggests that binding of the KRAS GTPase domain to BRAF is not an energetic driver of activating rearrangements.” This is a confusing statement, as the similar affinity of KRAS for the autoinhibited and active states of full-length BRAF cannot lead to that conclusion that “binding of the KRAS GTPase domain to BRAF is not an energetic driver of activating rearrangements”. Furthermore, this comment is overestimating the results obtained in an invitro settings in the absence of the membrane, and the full range on inhibitory interactions, this part of the text and conclusions should be revised according to the experimental settings and reflect more on the physiological interpretation of such results using isolated domains of the proteins.

7. To compare the activity of the autoinhibited complex with or without addition of KRAS and/or liposomes, please at least show the autoinhibited complex run in each gel so that the readers can really see if there is a change or not. The comparison between different blots is problematic. Complete membrane images must be included as a supplementary figure, to enable interested readers to properly analyze blots.

8. The major activation step proposed in this study is that there is a dynamic conformational equilibrium between the “recruitment complex” and a hypothetical “open monomer” complex. The existence of such an “open monomer” complex and its co-existence with a “recruitment complex” need to be experimentally validated. Can the authors propose mutations that would stabilize the ‘open monomer’ complex and/or alter the dynamic equilibrium, to allow the identification of the complex biochemically/biophysically. Furthermore, if in the model proposed by the authors the initial binding with RAS does not drive the conformational changes required for disrupting the autoinhibited monomer, where does the energy come from to dislodge the CRD? This point should also be discussed in the manuscript.

Minor Comments:

- The authors use “Ras”, “RAS” and “Raf”, “RAF” through the manuscript. Please be consistent.
- Please include molecular weight markers for the Western blot membrane images in figure 4. Likewise, please include an extended/supplementary figure with complete, uncropped Western blot membrane images.

- In the legend for Figure 3, please describe what the error bars represent (i.e. standard deviation, standard error of the mean). Ideally, the table in Figure 3 should be shown as a table, not a figure panel.

Detailed response to Reviewers' comments:

We thank the referees for their overall enthusiasm for our work and their insightful suggestions and questions. We have worked to address their concerns in full, as described below. Reviewer comments are presented in full in black text, our responses are in blue.

Reviewer #1 (Remarks to the Author):

The manuscript entitled 'Cryo-EM structure of a RAS/RAF recruitment complex' by Park et al. describes cryo-EM reconstructions of KRAS bound to intact BRAF in an autoinhibited state with MEK1 and a 14-3-3 dimer. The authors solve reconstructions of the complex and find the Ras-binding domain (RBD) of BRAF in two different orientations. Given the reduced density/lower resolution for both KRAS and BRAF RBD relative to the rest of the complex, the authors postulate conformational variability of the RBD. These findings are in contrast to a recent Cryo-EM structure solved by Martinez Fiesco et al (2022) in which the RBD forms contacts with 14-3-3 in an apparently more fixed conformation, suggesting that RAS interactions with 14-3-3 may obscure binding to the RBD.

Herein, the investigators postulate that BRAF autoinhibitory contacts are unperturbed by RAS binding to the RBD given the conformational dynamic nature of the RBD. To obtain further support for their hypothesis, the authors conduct a series of in vitro binding measurements and activation assays. Consistent with their Cryo-EM data, they find similar affinity of RAS for RAF between the autoinhibited and active BRAF complexes with significant BRAF activity observed only in the context of processed KRAS in the presence of liposomes. Hence, they propose that when RAS engages the RBD, the CRD autoinhibitory contacts with 14-3-3 and the kinase domain remain intact (intermediate state) and that RAS binding alone is not sufficient to activate the CRAF.

The cryo-EM reconstructions of KRAS bound to intact autoinhibited BRAF described in this manuscript, lend new insights into what the investigators term a BRAF 'pre-activation intermediate'. Moreover, in conjunction with RAS binding and RAF activation assays, their findings support previous work in the field that RAS binding to the RAF-RBD localizes cytosolic RAF to the membrane but additional steps are required to promote activation. However, some concerns exist with the manuscript in its present state.

As some of the conclusions in current study are based on indirect readouts and the Cryo-EM reconstruction solved in the absence of the membrane, the manuscript would be strengthened by summarizing/referencing the previous body of work that RAS binding alone is not sufficient to activate RAF to better lay the foundation for their hypothesized intermediate.

We have added a short paragraph to the introduction to summarize (and emphasize) this important background information.

Moreover, error analysis is missing for the binding measurements and the activity measurements lack quantification.

We now provide standard deviations for the binding measurements and we have added quantitation for the western blots in the activation experiments. See below for additional details.

The manuscript would also be strengthened by the inclusion of more detailed information regarding RAS/RBD and CRD autoinhibitory interactions in the reconstructions.

We have added text to make clear that the KRAS/RBD interactions are essentially the same as observed in prior crystal structures of KRAS/RBD complexes. Given the limited resolution in this portion of the structure, we do not feel it appropriate to describe specific residue by residue interactions. We have also added a supplemental figure (S3) to illustrate the position of the RBD domain in the present structures as compared with prior autoinhibited-state structures. The CRD domain and its interactions are also unchanged as compared with our prior autoinhibited-state structure. See below for our text edits regarding this point.

Mutational analyses (binding/activation assays) of the previously proposed autoinhibitory RAF RBD/14-3-3 contacts may add additional support for the proposed intermediate state and mechanistic interpretation of the results.

Martinez Fiesco et al. conducted an extensive mutational analysis of the RBD/14-3-3 interface they observed in their structure (Martinez Fiesco et al., Nature Communications 2022). They found modest but reproducible changes in BRET signal between tagged regulatory and catalytic fragments of BRAF with a series of substitutions in the RBD domain (from a ~17% increase to ~10% decrease), and also found an increase in focus formation by NIH-3T3 cells transfected with a mutant that decreased BRET signal. We feel these results speak for themselves with respect to the role of the RBD in autoinhibition. Also, given our finding that the autoinhibited state is maintained upon binding of KRAS (and that these interactions are absent in complex with KRAS), the predicted effect of these mutations would pertain primarily to the basal level of activity of the WT vs. mutant autoinhibited state, and we are not confident that we could reliably measure such small differences in specific activity between different protein preparations.

More detailed questions/concerns/suggestions are listed below.

There is a large body of data on RAS mediated activation of RAF. Of note, RAS binding to the RAF RBD has been shown to promote membrane association and is a key step yet insufficient for RAF activation. These previous findings should be concisely summarized to lay the foundation for the proposed pre-activation intermediate.

As noted above, we have added a short paragraph to the introduction to summarize this important background information.

Is the RAS-RAF RBD binding interface retained in the 'preactivation intermediate' (e.g. similar to that determined from previous structural studies)?

Yes. It is essentially identical in the "Ras-up" structure (relative orientations of the domains is the same, despite the fact that prior crystal structures are CRAF and present structure is BRAF) We have revised the text to make this clearer:

The interaction between KRAS and the BRAF RBD in this structure is essentially the same as that previously seen in crystal structures of CRAF bound to RAS^{15,17}; KRAS uses primarily its "switch I" region to bind the RBD, forming an anti-parallel β -strand interaction with the β 2-strand of the RBD domain (Supplementary Fig. 2a,b).

In the Ras-front structure, there is an $\sim 11^\circ$ difference in relative orientation of the domains, but the interface is preserved. Similar angular differences have been noted in comparing different crystal structures of RAS/RBD complexes.

Additional information clarifying the nature of the RAS-CRD autoinhibitory contacts and whether these contacts are at all altered upon RAS engagement is needed.

This reviewer might have noticed small differences in the structure of the CRD domain in the present structures as compared with our previous autoinhibited structure (PDB 6NYB). These differences are a result of an improved model in this region which was enabled by 1) reference to an Alphafold2 model of the BRAF CRD and 2) improved resolution in this region in the current structures. Looking back at our prior maps, the newly modeled CRD clearly improves the fit in that map as well. In short, there is no discernable difference in the CRD autoinhibitory contacts with/without KRAS. We have added a short section to the text to make this point:

Comparison of the present RAS-up and RAS-front structures with our previously determined RAS-free autoinhibited structure reveals that the constellation of autoinhibitory interactions among the CRD domain, C-lobe of the kinase domain, and the 14-3-3 dimer and its cognate phospho-serine recognition sites is unchanged. Minor differences in the CRD domain in the present structures do not reflect differences due to binding of KRAS. Rather they are a result of an improved fit to the cryo EM maps enabled by reference to an Alphafold2 model of this domain and modestly improved resolution in this region for the RAS-bound structures.

The resolution of KRAS in non-crosslinked and BS3-crosslinked cryo-EM maps is low (8 - 9.5 Å) and limits analysis of the data.

While the resolution is low, it does allow us to confidently position both KRAS and the RBD domain. We now provide Supplemental Videos 1 & 2 to illustrate the fit to the maps in this region in the KRAS-up and KRAS-front structures. Prior crystal structures of KRAS/RBD complexes define details of this interactions. See also response to Reviewer 3, point 4, below.

The authors evaluate the binding affinity of RAS with RAF but should compare their findings to previous reported data (some on CRAF). Errors are lacking in the Table associated with Figure 3.

We have moved the table portion of prior Figure 3 to a free-standing table (now Table 1) and added comparisons with previous reported binding data for both BRAF and CRAF. Additionally, we now provide standard deviations for our binding measurements. The binding curves used to derive the values in the table are now found in Supplementary Figure S6. See also our response to Reviewer 2, point 4, below.

The activation assays are not quantified so that a relative comparison can be made.

We have now quantitated the western blot data and incorporated the integrated western blot band intensities in the Figure (now Figure 3.)

Reviewer #2 (Remarks to the Author):

In the manuscript by Park et al., new structural insights about RAS/RAF/MEK/ERK signaling cascade are shown. The importance of this signaling cascade in cancer development is well-documented, but despite all the effort to understand details surrounding this cascade, some important questions are still left unanswered. In this study, authors addressed one of those remaining questions: the effect of RAS binding on the downstream signaling complex composed of RAF, MEK1, and 14-3-3 proteins. In addition, the authors probed whether this is the crucial step in transition of BRAF from an autoinhibited monomeric state to an activated dimeric state. Two cryo-EM structures of RAS/BRAF/MEK/14-3-3 complex, exhibiting the difference in the orientation of RAS protein relative to the rest of the complex (“RAS-up” and “RAS-front” conformations), are reported. The authors provide the structural evidence that BRAF remains in autoinhibited state upon RAS binding, suggesting that the obtained complex represents a pre-activation intermediate in autoinhibitory state. Interestingly, in their biochemical assays, the authors managed to observe improved BRAF activation upon addition of fully modified KRAS (KRAS4b-FME) and phosphatidylserine-containing liposomes (PS-liposomes). Overall, the manuscript is nicely written and organized and is worthy of publication in Nature Communications. However, before publishing, the authors need to address a few major and minor concerns.

Major concerns:

1. In the reported cryo-EM structures, the authors note conformational variability regarding the RAS positioning. However, the analysis of this structural variability is missing. To address this, the authors should consider performing additional steps in cryo-EM processing (either multi-body refinement or 3D variability analysis) of their data and present the analysis in form of a movie.

We have analyzed the 3D variability in our non-crosslinked structure and prepared a movie to illustrate it. The movie is incorporated as Supplementary Video 3. As the movie shows, density for the KRAS/RBD region largely fades out between the “up” and “front” positions, indicating a lack of particles with intermediate conformations.

2. Along the same lines, on page 5 the authors report inconsistency between reported structures and obtained mass spectrometry data of crosslinks. After completing the suggested 3D variability analysis, it would be interesting to see if the crosslink mass spec data align with the structural flexibility.

No, neither of these crosslinks appear to be consistent with positions intermediate between RAS-up and RAS-front.

3. The authors state on page 4 that the “RAS-front” conformation is not induced by cross-linking, since they can visualize the same conformation in their non-crosslinked sample (Supplementary Fig.1a). However, looking at the provided data processing flowcharts (Supplementary Figure 4 and Figure 5), it is difficult to discern which maps represent the “RAS-front” conformation. Moreover, it is unclear how many particles from the entire particle stack contributed to that conformation/state.

Hence, to aid the reader, “RAS-up” and “RAS-front” maps should be colored differently, the number of particles contributing to both conformations/states should be clearly shown and maps should be highlighted in both non-crosslinked and crosslinked processing schematics (Supplementary Figure 4 and Supplementary Figure 5).

We have incorporated the coloring scheme and added particle numbers as suggested to these Supplementary Figures (Now Supplementary Figures S7 and S8).

4. In Figure 3, except for the BRS-RBD-CRD plot, it appears that the authors are consistently missing the higher concentration points, suggesting that full saturation of binding sites was not attained under the assay conditions used. This could impact the K_d value estimate and additional titration points should be provided.

We re-analyzed our MST data to plot “fraction bound” for each BRAF fragment or complex. All samples reach 85% or greater saturation, and the fraction-bound analysis makes clear that with the exception of the RBD-only construct, there is not a significant difference in affinity for KRAS. We have added this analysis to the binding curves, which are now presented in Supplementary Figure S6 (previously part of Figure 3 in the main text). We are unable to achieve higher concentrations for these samples (because higher concentrations lead to aggregation), but it is clear from the fraction bound curves that this is unlikely to have a significant effect on measured affinities (which also agree well with literature values).

5. Equally importantly, have the authors assessed the integrity of each protein in the MST buffer before performing MST experiments? If so, what are the results? If not, then such tests should be performed. This is particularly valuable for binding assays completed under higher concentrations of a binding partner.

We have. The buffer used for the MST experiments mimicked our size-exclusion buffer, but with the addition of 0.05% Tween-20. Addition of Tween-20 to our samples did not affect their solubility to the highest concentrations employed. We have added a line to the methods section for MST to clarify this.

Minor concern:

- Figure 1c, right panel – authors should clarify if the lane 3 contains higher amount of the sample loaded, and if so, by how much.

Lanes 2 and 3 represent the sample loaded before and after ~7-fold concentration, respectively. We have edited the figure and legend to clarify this point.

Reviewer #3 (Remarks to the Author):

In this manuscript by Park, Rawson and colleagues, the group reports cryo-EM structures of a KRAS bounding to BRAF/MEK/14-3-3(2) autoinhibited complex purified from baculovirus-infected insect cells. The KRAS/BRAF/MEK1/14-3-3 complex shows KRAS bound to the flexible RAS-binding domain (RBD) of BRAF in two orientations.

In addition, the authors carried out in vitro biochemical activation studies which confirmed that membrane recruitment is needed for BRAF activation, consistent with previous studies. The structures of RAS-BRAF complexes are of general interest to those in the kinase and RAS/RAF signaling field. The results of this study may be suitable for publication in Nature Communications, once the following major concerns have been addressed.

Major comments:

1. The BRAF used in this study is human BRAF expressed and purified from a heterologous (insect) expression system. Differences in post-translational processing in the insect cell expression system

(particularly phosphorylation) may differ between the insect-cell expressed protein and the endogenous human protein. This point needs to be discussed in the manuscript.

We now discuss this point explicitly in the manuscript, including in the context of S151 phosphorylation and potential effects on the positioning of the RBD, as further discussed below.

Moreover, can the authors analyze the phosphorylation state of the BRAF protein used in this study and compare it to that expressed from a mammalian expression system? In particular, it is known that the Ser151 of BRAF is maintained in a phosphorylated state due to the high level of basal ERK activation in Sf9 insect cells. As Ser151 is in close vicinity of the RBD, phosphorylation of S151 could directly interfere with the RBD region and Ras binding to BRAF complexes purified from an insect expression system. Have the authors investigated this issue? This point should also be discussed in the manuscript.

This is an interesting point which we have now further investigated in depth. Our Sf9-expressed BRAF/MEK/14-3-3 protein is indeed highly phosphorylated on S151 of BRAF – approaching full site-occupancy. We also found this site to be phosphorylated in our BRAF/14-3-3 complex prepared in mammalian Expi293f cells (although we were not able to quantitate it). This is a well-known and widely observed phosphorylation site, with a putative role in feedback regulation of Raf. Ser151 phosphorylation is certainly not unique to Sf9 cells, and the extent of phosphorylation on this site could be influenced by a variety of factors beyond choice of expression system, including cell culture and purification conditions. Martinez Fiesco et al. do not report the phosphorylation state of this site in their work (in which they define a position of the RBD in the autoinhibited state).

To address the potential effect of S151 phosphorylation on the order and interactions of the RBD, we prepared an autoinhibited complex with a truncated BRAF construct lacking the first 155 residues (which are not ordered in either structure). Cryo EM imaging and single particle reconstruction with this $\Delta 155$ -BRAF/MEK1/14-3-3 complex to 4.1 Å resolution yielded a map that was essentially the same as that obtained with the full-length BRAF complex, including weak, ill-defined density in the region of the RBD. Considering this result, and that the Ser151 region is not resolved in the Martinez Fiesco et al. structure, it seems quite unlikely that Ser151 affects the position/interactions of the RBD in the autoinhibited state, irrespective of its phosphorylation. The $\Delta 155$ -BRAF/MEK1/14-3-3 cryo EM map is presented in Supplementary Figure S5A, and is shown at two contour levels fit with our prior autoinhibited structure (PDB 6NYB) and the Martinez Fiesco autoinhibited structure (PDB 7MFD).

We also sought to more broadly address the role of the RBD in stabilizing the autoinhibited state by preparing an autoinhibited complex with a truncated BRAF construct lacking the RBD domain (Δ RBD-BRAF, residues 233-766) of BRAF. We were able to prepare apparently autoinhibited complexes with Δ RBD-BRAF/MEK1/14-3-3 (the ternary complex eluted on SEC as expected for the autoinhibited complex) and confirm with cryo EM imaging that it indeed adopts the autoinhibited conformation seen in the full-length and $\Delta 155$ -BRAF structures. Although we were unable to readily obtain 3D reconstructions with this dataset, the most abundant class average was the stereotypical “front view” of the autoinhibited complex. We conclude from this – consistent also with our KRAS-bound structures – that interactions of the RBD are not *required* to maintain the autoinhibited state. That is not to say that they may not contribute to its stability, consistent with the observations of Martinez Fiesco. We do note a subjective increase in 2D classes that appear to represent “broken” or partially

open autoinhibited particles, perhaps with the CRD released (see Supplementary Fig. S5B in revised manuscript).

2. The system and methodology used in the current study to produce the BRAF autoinhibited complex used to study binding to RAS was previously reported by the same group in Park et al (Nature, 2019), yielding a BRAF autoinhibited structure with an insufficient resolution of the RBD to model the position and orientation of the RBD, indicating it was already released from the autoinhibited complex, flexible, and solvent exposed for RAS binding. In contrast, the RBD domain in the full length BRAF/MEK1/14-3-3 and BRAF/14-3-3 autoinhibited complexes from mammalian cells was previously resolved in Martinez Fiesco et al (Nature Communications, 2022). The authors found that the RBD of BRAF is partially occluded by 14-3-3, requiring structural rearrangements to permit RAS to fully bind to BRAF, which was further validated using mutagenesis experiments. The working system differences as well as the different autoinhibitory complexes should also be discussed in the manuscript.

We've largely addressed this question in our response to point 1 above. But, in further response: Supplementary Figure S5A in the revised text shows the Martinez Fiesco structure (PDB ID 7MFD) placed in our current $\Delta 155$ -BRAF/MEK1/14-3-3 map, which shows weak density corresponding to the RBD in approximately the position modeled in 7MFD. (We also observed weak density in this position in our prior autoinhibited structure).

Martinez Fiesco et al (Nature Communications, 2019) proposed a model for how the monomer to dimer transition in BRAF is mediated by RAS binding in presence of membrane. The monomer to dimer transition model is in the context of membrane presence instead of RAS binding alone and is misstated in several places in this manuscript. Please correct.

Thank you for pointing this out. We didn't intend to imply otherwise and have now revised the text to explicitly state that their model is in a membrane context.

3. Similarly, as mentioned before the approximate position of the RBD in the BRAF/MEK1/14-3-3 complex was previously reported in Park et al (Nature, 2019) by the same group, but the resolution was not sufficient to position the known RBD model. Can the authors provide analysis of the RBD region before and after KRAS binding? Are the two orientations of RBD in the KRAS/BRAF/MEK1/14-3-3 complex also observed in the BRAF/MEK1/14-3-3 complex?

The KRAS-bound orientations of the RBD differ from the position of the RBD in the autoinhibited state. This is now illustrated in Supplementary Figure S3 of the revised manuscript.

4. The novelty of the reported structure is the interaction between RAS and the autoinhibited RAF complex. Unfortunately, the resolution at the RAS-RAF RBD region is very poor (worse than 8Å), lacking well-defined secondary structure, preventing the authors from reporting details of interactions. We strongly recommend the authors to get better quality data for this part of the structure that will allow them to accurately model the RBD domain and RAS protein in their map and draw conclusions.

We tried very hard to improve the resolution in this region, including attempting focused refinement on the KRAS/RBD module. Not surprisingly, this region proved to be too small for this to be

successful. Of course, the relatively low resolution of the KRAS/RBD is a result of (and evidence of) its relative mobility, which is one of our conclusions.

Fortunately, there is a high-resolution crystal structure available for KRAS bound to the CRAF RBD, as well as other similar structures, including with other RAS isoforms. Thus the details of the KRAS/RBD interaction are already well defined. We can unambiguously place both KRAS and the BRAF RBD (homology modeled from the CRAF crystal structure) in our map, and their positions recapitulate the known mode of interaction of these domains.

With new improved maps, can the authors provide semi-transparent map density of the KRAS and RBD regions to help the reader evaluate this interface? Are the KRAS and RBD regions well resolved? What is the local resolution of the critical KRAS and RBD regions? Please include a supplementary figure with details of the cryo-EM density for this interface so the reader can judge if the map allows identification of the KRAS and RBD region.

As noted above, the position of KRAS and the RBD are sufficiently defined in our maps to allow us to confidently model them. We have added Supplementary videos to illustrate the density and model in this region for the KRAS-up (Supplementary Video 1) and KRAS-front structures (Supplementary Video 2).

5. The binding curves for the different BRAF constructs with KRAS seems not to reach saturation what could lead to overestimation of the KD values, we recommend achieving saturation for its determination.

See response to Reviewer 2, point 4, above.

In addition, please give an estimation of the error for the affinities determined using MST. Please include these values in a table. Please provide a supplementary figure with the raw MST traces. Please explicitly state in the methods how many times the experiment was performed. Ideally, MST experiments should be performed three times and the affinities averaged to give a reliable estimation of the affinity value and the error.

As noted above, error estimates are now provided in the Table (now Table 1), and representative MST traces are moved to Supplementary Figure 6. The experiment was performed three times and averaged to derive the reported affinities and error estimates. We now note this in the methods.

6. On page 6, line 189, the authors state “Furthermore, the similar affinity of KRAS for the autoinhibited and active states of full-length BRAF suggests that binding of the KRAS GTPase domain to BRAF is not an energetic driver of activating rearrangements.” This is a confusing statement, as the similar affinity of KRAS for the autoinhibited and active states of full-length BRAF cannot lead to that conclusion that “binding of the KRAS GTPase domain to BRAF is not an energetic driver of activating rearrangements”. Furthermore, this comment is overestimating the results obtained in an invitro settings in the absence of the membrane, and the full range on inhibitory interactions, this part of the text and conclusions should be revised according to the experimental settings and reflect more on the physiological interpretation of such results using isolated domains of the proteins.

We deleted this confusing (and perhaps over-reaching) statement.

7. To compare the activity of the autoinhibited complex with or without addition of KRAS and/or liposomes, please at least show the autoinhibited complex run in each gel so that the readers can really see if there is a change or not.

There is far too little of the autoinhibited complex to be seen with Coomassie staining. Stripping and re-probing the membranes with an anti-BRAF antibody was not successful. However, the anti-ERK1/2 blots essentially serve as a loading control for the BRAF complex as well since they were dispensed as a “master mix” that contained both.

The comparison between different blots is problematic. Complete membrane images must be included as a supplementary figure, to enable interested readers to properly analyze blots.

We have provided complete membrane images as Supplementary Data File 3. There is no issue with comparing between blots, as all panels for each experiment were in fact crops from the same blots.

8. The major activation step proposed in this study is that there is a dynamic conformational equilibrium between the “recruitment complex” and a hypothetical “open monomer” complex. The existence of such an “open monomer” complex and its co-existence with a “recruitment complex” need to be experimentally validated. Can the authors propose mutations that would stabilize the ‘open monomer’ complex and/or alter the dynamic equilibrium, to allow the identification of the complex biochemically/biophysically.

We have previously done this with the S365A mutation. This mutation ablates one of the two 14-3-3 binding sites required for autoinhibition and yields a mix of open monomers and active dimers. The highest peak in a complex prepared with BRAF-S365A is at the volume expected for a monomer (though a smidge earlier that observed for the WT, fully autoinhibited complex). Extended Data Fig. 7a from our Park et al. Nature 2019 paper is reproduced below. One could worry that the open monomers were active rather than fully inhibited (based on phosphorylation of exogenous MEK substrate as shown in blot above the gel), but this activity is more likely due to subsequent partitioning of the open monomers into active dimers. (Dimers elute in a broad peak at around 12-13 mls, the monomer complexes at ~14 ml.)

While this shows that the open monomer can be isolated and is not hypothetical (as an aside, how could one transition between autoinhibited monomer and active dimer, if not through an open monomer?), it does not provide evidence of a dynamic conformational equilibrium between the “recruitment complex” and the “open monomer” state. In our Discussion section, we have presented this proposed equilibrium as a model – consistent with available data – that can explain why

dephosphorylation of pS365 matters for RAF activation. To further emphasize that this is a model and not an experimental fact, we have revised the wording of the text and added a “?” below the arrows indicating a reversible equilibrium between the recruitment complex and open monomer in Figure 4. We think this is a fair treatment, and an important piece of our manuscript that integrates the present work with recent work on the SHOC2 complex.

Furthermore, if in the model proposed by the authors the initial binding with RAS does not drive the conformational changes required for disrupting the autoinhibited monomer, where does the energy come from to dislodge the CRD? This point should also be discussed in the manuscript.

We have expanded on this point in the discussion as follows:

An important outstanding question is how “opening” of the autoinhibited complex is promoted in the membrane environment. It is clear from the present work that steric effects resulting from binding to KRAS are not sufficient, although they may contribute to the extent that they disrupt stabilizing interactions of the RBD. We can conceive of two general mechanisms that could contribute to opening. In one, physical interactions of the recruitment complex with the membrane could directly induce its opening, whether by steric, interfacial effects, or via specific interactions. In the second, proximity to the membrane could promote association of the CRD with KRAS and the membrane upon “breathing” (transient opening or partial opening) of the complex, thereby increasing the fraction of time in the open state.

We have also revised the paragraph on the role of the SHOC2 phosphatase complex.

Minor Comments:

- The authors use “Ras”, “RAS” and “Raf”, “RAF” through the manuscript. Please be consistent.

We have standardized to the uppercase RAS and RAF. We use BRAF when referring to BRAF specifically, and RAF when generalizing to other isoforms.

- Please include molecular weight markers for the Western blot membrane images in figure 4. Likewise, please include an extended/supplementary figure with complete, uncropped Western blot membrane images.

We have included Supplementary Data File 3 with the full uncropped Western blot membrane images, including the labeled molecular weight markers.

- In the legend for Figure 3, please describe what the error bars represent (i.e. standard deviation, standard error of the mean). Ideally, the table in Figure 3 should be shown as a table, not a figure panel.

As requested, the Table in former Figure 3 is now shown as Table 1, with error estimates provided. We now explain in the legend for these traces (Supplementary Figure S6) that the error bars represent standard deviations.

REVIEWER COMMENTS

Reviewer #2 (Remarks to the Author):

The authors have adequately addressed all the concerns raised in my initial review. They have made significant improvements to the manuscript, and I strongly recommend it for publication in Nature Communications.

Reviewer #3 (Remarks to the Author):

The revised manuscript has shown improvement compared to the initial submission. The authors have addressed most of the reviewers' suggestions. However, there are still some outstanding points that require further attention, especially in light of the new data presented in the rebuttal. These points are listed below and should be addressed before the manuscript can be accepted for publication.

1. One major concern raised in the previous revision was the consequences of BRAF post-translational modification resulting from the use of an insect cell expression system in the manuscript. Specifically, the concern was regarding the phosphorylation of Ser151 on the RBD domain and how it may differ between the protein expressed in insect cells and the BRAF in human cells.

In response to this question, the authors state that BRAF produced from insect cells is highly phosphorylated on Ser151. To investigate the effect of this phosphorylation on the RBD domain, the authors produced two N-terminal truncations of BRAF and prepared them in insect cells in a manner similar to the full-length complex.

The first truncation, Δ 155-BRAF, lacks the first 154 residues and exhibits the overall autoinhibited conformation as reported by Park et al. and Martinez Fiesco et al. However, importantly, the RBD (even at low resolution) adopts the position reported by Martinez Fiesco et al., as shown in Figure S5a, rather than the 'up' or 'front' conformation observed when the protein is expressed in insect cells and highly phosphorylated on Ser151, as reported in this manuscript. This clear evidence indicates that the RBD position and its potential interactions are highly influenced by the posttranslational modification in the insect cell expression system used. This information should also be indicated in Figure S3. The authors acknowledged this observation in the rebuttal letter but not in the manuscript. It should be included and discussed in the manuscript as a potential drawback of the

expression system used, indicating that the observed 'up' or 'front' positions of the RBD are dependent on the posttranslational modifications occurring during the insect cell expression of BRAF, and they differ from the RBD position when BRAF is expressed in mammalian systems.

Furthermore, with the second truncation, Δ RBD-BRAF, where the RBD was completely removed, the authors showed an increase in cryoEM particle heterogeneity, including 2D classes in which the autoinhibited state is disrupted, displaying a released CRD (Figure S5b). This provides further indication of the role of the RBD and the importance of the RBD:14-3-3 interaction in maintaining the autoinhibited conformation, as reported by Martinez Fiesco et al. This should be discussed in the manuscript.

The conclusions and assertions made by the authors in the manuscript, such as "it is unlikely that phosphorylation of S151 affects interactions of RBD domain" or "Thus, the RBD is not required for the maintenance of the autoinhibited state," are inaccurate and contradict the data reported in the manuscript and the new data provided in the revision. These statements should be corrected to properly reflect the data before publication.

2. Considering that the reported position of the RBD ('up' or 'front') in this manuscript may not accurately reflect its position or orientation when BRAF is expressed in mammalian cells, the conclusion that "autoinhibitory interactions in the complex are unperturbed by the binding of KRAS" might be overstated. I suggest that the authors adopt a more nuanced approach in their conclusions. Ideally, this claim should be supported by experimental data, such as a cryoEM study of the KRAS/ Δ 155-BRAF complex. Investigating how Δ 155-BRAF, which more closely resembles mammalian BRAF, interacts with KRAS could provide informative insights.

3. There are certain flaws in the proposed RAF activation model that need to be addressed, or alternatively, the text should be modified to reflect a more nuanced understanding.

In my previous review, I requested experimental evidence for the dynamic conformational equilibrium between the "recruitment complex" and a hypothetical "open monomer" as a major activation step (step 2) in the proposed model. However, this remains unanswered. In response, the authors referred to previously published data on the S365A mutation in the BRAF/MEK/14-3-3 complex, which suggests a potential open monomer. This would indicate an equilibrium between the autoinhibited monomer and an open monomer, prior to step 1 in the proposed model, rather than an open monomer induced by RAS binding or post RAS binding (step 2). Thus, the second step is still speculative.

Additionally, I believe that a shift in elution position in preparative size exclusion chromatography alone is not strong evidence. Ideally, a technique that directly measures molecular mass, such as multi-angle light scattering coupled to SEC, analytical ultracentrifugation, or mass photometry, would be more suitable. However, I understand that performing these experiments is not requested at this time.

Moreover, the authors concluded that "steric effects resulting from binding KRAS are not sufficient for the opening of the autoinhibited state." However, considering that the RBD position in the insect-cell expressed protein does not reflect the same position as in the mammalian expressed protein, this statement should be modified to reflect the system used in this study.

Minor points:

- The blots in the main Figure 3 still lack molecular weight markers.
- It is still unclear whether the MST data represents replicates within a single experiment or the average of three independent experiments. To clarify, were the same dilution series measured three times, or was the dilution series prepared and measured three times? Additionally, the authors should comment on the large errors on the KD determinations and what that implies for the analysis.

Reviewer #4 (Remarks to the Author):

In their manuscript "Cryo-EM structure of a RAS/RAF recruitment complex" Park et al describe the in vitro reconstructions of KRAS bound to intact BRAF in an autoinhibited state together with MEK1 and 14-3-3. Of interest, the authors observed a reduced resolution for the RAS-RAF RBD region indicating a relative mobility in that region, but can confirm interactions by crosslinking mass spectrometry (XL-MS).

The manuscript is well written and of great value for the readership of Nature Communications. The authors already gave detailed responses to the questions of previous reviewers. Hence, this reviewers' comments will focus on the quality of the crosslinking data, which is also in line with the reviewers expertise.

The postulated conformational variability of RBD seems very exciting and novel to me, especially as it seems to be in contrast to the previously published structure by Fiesco in the context of membrane presence. Previous reviewers already suggested that the low resolution in the RAS-RAF RBD region prevents from reporting details on interaction and the authors responded that this region was too small to improve resolution. XL-MS data might be an optimal strategy useable to remodel the interaction surface and to confirm interaction. To that end I have several comments:

-Can you use your XL-MS data to do that? It seems from the supplementary dataset that no RBD to RAF or RAS XL was found and no link at all is reported on KRAS, but this is in contrast to Figure S4 where such links are plotted.

-If you do not have sufficient links, the conclusions from XL-MS data would very much profit (if feasible) to be supplemented with XL data with a second linker that does not target Lys residues. E.g. SDA (Sulfo-NHS-Diazirine) is a heterobifunctional linker that would target all amino-acids after photoactivation increasing chances to confirm interactions within the investigated protein complex. Another option would be to couple carboxylic acids (i.e. D,E) to primary amines (i.e. K) using a coupling agent as DMPTMM which would be a zero length linker enabling highest possible resolution from XL-MS data.

-Since the BS3 links seem to be inconsistent to the CryoEM data suggesting flexibility (as also mentioned by Reviewer 2 in the previous revision), would it be possible to re-model the structure to obtain another fitting confirmation giving an idea to the other conformations existing? Tools like I-TASSER (iterative threading assembly refinement) or HADDOCK (high-ambiguity-driven protein-protein docking) have been used for this task previously, but there might be tons of others around as well.

-In Supplemental Figure 4, you show identified BS3 XLs: As mentioned before please check why this does not align with links reported in the supplemental data-set where I can only see interdomain links from 14-3-3 to BRAF and Intralinks but no other inter-domain link.

-In the same Figure the authors show which links do not fit to the maximum expected distance of 20Å. Can you elaborate in more detail how you confirmed those links are still confident (i.e. based on which FDR control was applied, why was a score difference of 15 for protein prospector chosen as sufficient – is this an empirical value?

-The authors describe, they manually validated their XLED spectra and provide a PDF showing all annotated spectra. I agree from my experience that those spectra look like confident matches to me as well. I am convinced though that a more detailed description of selection criteria/ chosen QC strategy /FDR threshold etc. would improve the quality of the manuscript. Especially the rather small database used for searching is potentially susceptible for false positive hits. Comparison to searches including some contaminants in the database as well, where no XL is expected to, could further improve confidence in the obtained results.

- Although the focus of this manuscript is on CryoEM, significant conclusions are driven from MS data as well which is why I would ask the authors to upload their XL-MS raw files, FASTA files, and ideally result files to a repository (e.g. MassIVE <https://massive.ucsd.edu/ProteoSAFe/static/massive.jsp> , PRIDE <https://www.ebi.ac.uk/pride/archive/> , or others are commonly used in the proteomics community) to enable full transparency and reanalysis by others.

Minor

-Although this is likely not influenceable by the authors, it is noted here, that the given link to ProteinProsector does not work at the time of the review. The referenced paper (<https://www.ncbi.nlm.nih.gov/pmc/articles/PMC2596346/>) for ProteinProspector does not explain how this software performs XL searches but references to an ASMS paper instead (again with a broken link to the Prospector website). Hence, assessing details of functionality for that software was not possible for the reviewer.

Reviewer comments are presented verbatim in black text, our responses are interspersed in blue text.

REVIEWER COMMENTS

Reviewer #2 (Remarks to the Author):

The authors have adequately addressed all the concerns raised in my initial review. They have made significant improvements to the manuscript, and I strongly recommend it for publication in Nature Communications.

Reviewer #3 (Remarks to the Author):

The revised manuscript has shown improvement compared to the initial submission. The authors have addressed most of the reviewers' suggestions. However, there are still some outstanding points that require further attention, especially in light of the new data presented in the rebuttal. These points are listed below and should be addressed before the manuscript can be accepted for publication.

1. One major concern raised in the previous revision was the consequences of BRAF post-translational modification resulting from the use of an insect cell expression system in the manuscript. Specifically, the concern was regarding the phosphorylation of Ser151 on the RBD domain and how it may differ between the protein expressed in insect cells and the BRAF in human cells.

In response to this question, the authors state that BRAF produced from insect cells is highly phosphorylated on Ser151. To investigate the effect of this phosphorylation on the RBD domain, the authors produced two N-terminal truncations of BRAF and prepared them in insect cells in a manner similar to the full-length complex.

The first truncation, Δ 155-BRAF, lacks the first 154 residues and exhibits the overall autoinhibited conformation as reported by Park et al. and Martinez Fiesco et al. However, importantly, the RBD (even at low resolution) adopts the position reported by Martinez Fiesco et al., as shown in Figure S5a, rather than the 'up' or 'front' conformation observed when the protein is expressed in insect cells and highly phosphorylated on Ser151, as reported in this manuscript. This clear evidence indicates that the RBD position and its potential interactions are highly influenced by the posttranslational modification in the insect cell expression system used. This information should also be indicated in Figure S3. The authors acknowledged this observation in the rebuttal letter but not in the manuscript. It should be included and discussed in the manuscript as a potential drawback of the expression system used, indicating that the observed 'up' or 'front' positions of the RBD are dependent on the posttranslational modifications occurring during the insect cell expression of BRAF, and they differ from the RBD position when BRAF is expressed in mammalian systems.

The reviewer appears to have missed a key point – we observe similar weak density for the RBD domain in both our full-length autoinhibited structure (as described in Park et al. Nature 2019, PDB entry 6NYB) and in the new Δ 155-BRAF structure reported here. Although we mentioned this in the revised manuscript, our brevity may have compromised clarity:

Single particle cryo EM reconstructions of Δ 155-BRAF/MEK1/14-3-3 complexes reveal an overall conformation essentially the same as that we previously observed for full-length BRAF complexes,

including weak density in the region of the RBD that was insufficiently resolved to allow modeling of this domain (Supplementary Fig. S5a). Thus it is unlikely that phosphorylation of S151 affects interactions of RBD domain.

In this revision, we have further emphasized this finding by including an additional panel in Supplementary Figure S5 that shows our previous cryo EM map obtained with full-length BRAF, allowing direct comparison with the $\Delta 155$ -BRAF map and the position of the RBD in the Martinez Fiesco structure. Comparison of the new panel with full-length BRAF (in which Ser151 is phosphorylated, Supplementary Figure S5b) with the $\Delta 155$ -BRAF in which this region is truncated (Supplementary Figure S5a) shows similar density in the region in which the RBD is modeled in the Martinez Fiesco structure. The phosphorylation state of Ser151 in the mammalian-expressed BRAF complex studied by Martinez Fiesco is not analyzed/reported in that manuscript, but irrespective of its phosphorylation state, this region is also disordered in their structure and does not appear to contribute to the observed interactions of the RBD.

Accordingly, we have now revised the manuscript text above to read:

Single particle cryo EM reconstruction of the $\Delta 155$ -BRAF/MEK1/14-3-3 complex reveals an overall conformation essentially the same as that we previously observed for full-length BRAF complexes (Supplementary Fig. S5a). In both this map and in our previously reported cryo EM map with full-length BRAF, we observe additional weaker density in the region of the RBD that is insufficiently resolved to allow modeling of this domain (compare Supplementary Figs. S5a, S5b). This density roughly corresponds to the position of the RBD as modeled in the Martinez Fiesco structure. These authors do not report the phosphorylation state of Ser151 in the mammalian-expressed BRAF used in their study, but irrespective of its phosphorylation state, this residue is also disordered in their structure and does not appear to contribute to autoinhibitory interactions of the RBD. Thus, available structural evidence does not support a role for phosphorylation of Ser 151 in modulating the autoinhibitory interactions of RBD domain.

Finally, the reviewer asserts that the “up” and “front” positions of the RBD that we observe in complex with KRAS are dependent on insect-cell specific posttranslational modifications, and may be suggesting that the difference in RBD position as compared with that observed by Martinez Fiesco et al. relates to expression system rather than to binding of KRAS. We do not agree. The same insect-cell expressed BRAF complexes were used in the present study in complex with KRAS as in our earlier study where we observe weak but not buildable density for the RBD in the same region as Martinez Fiesco et al. with mammalian-expressed BRAF. Thus the difference is clearly due to binding of KRAS. Secondly, there is no evidence that we are aware of confirming differences in PTMs in the N-terminal region of insect-cell vs. mammalian expressed BRAF. As noted above, our insect cell-derived protein is highly phosphorylated on Ser151, but we also find this modification in BRAF complexes prepared in mammalian cells (although we have not quantitated it). Finally, as Martinez Fiesco et al. point out in their manuscript, the position of the RBD that they observe would not allow binding of KRAS without creating steric clashes. Binding of KRAS was therefore expected to rearrange the RBD.

We very much appreciate the advance of Martinez Fiesco et al. in defining autoinhibitory interactions of the RBD that were not resolved in our earlier structure (Park et al. 2019). Both Martinez Fiesco et al. and the present reviewer suggest that the difference may be due to mammalian vs. insect cell expression. While this is certainly possible, it is just one of many differences in experimental conditions between the two studies. We have tested this reviewer’s suggestion that phosphorylation of Ser151 might explain the difference, but our findings do not support this idea.

Furthermore, with the second truncation, Δ RBD-BRAF, where the RBD was completely removed, the authors showed an increase in cryoEM particle heterogeneity, including 2D classes in which the autoinhibited state is disrupted, displaying a released CRD (Figure S5b). This provides further indication of the role of the RBD and the importance of the RBD:14-3-3 interaction in maintaining the autoinhibited conformation, as reported by Martinez Fiesco et al. This should be discussed in the manuscript.

We previously wrote:

The second truncation we studied removes the RBD domain as well (Δ RBD-BRAF, residues 233-766). While we were unable to obtain a 3D-reconstruction with this sample, the most abundant 2D class averages clearly show the configuration of the autoinhibited complex (Supplementary Fig. S5b). Thus the RBD is not required for maintenance of the autoinhibited state. That said, we do note a subjective increase in particle heterogeneity in this sample, including 2D classes in which the CRD domain appears to have been released from its 14-3-3 interactions (Supplementary Fig. S5b).

In light of the Reviewer's comments, we have now revised this section as follows:

The second truncation we studied removes the RBD domain as well (Δ RBD-BRAF, residues 233-766). While we were unable to obtain a 3D-reconstruction with this sample, the most abundant 2D class averages clearly show the configuration of the autoinhibited complex (Supplementary Fig. S5c). Thus, BRAF can adopt the autoinhibited state even in the absence of the RBD domain. However, in support of a role for the RBD in contributing to the stability of the autoinhibited state, we do note a subjective increase in particle heterogeneity in this sample, including 2D classes in which the CRD domain appears to have been released from its 14-3-3 interactions (Supplementary Fig. S5c).

The conclusions and assertions made by the authors in the manuscript, such as "it is unlikely that phosphorylation of S151 affects interactions of RBD domain" or "Thus, the RBD is not required for the maintenance of the autoinhibited state," are inaccurate and contradict the data reported in the manuscript and the new data provided in the revision. These statements should be corrected to properly reflect the data before publication.

We have removed both of these statements, as indicated in the revised passages above.

2. Considering that the reported position of the RBD ('up' or 'front') in this manuscript may not accurately reflect its position or orientation when BRAF is expressed in mammalian cells, the conclusion that "autoinhibitory interactions in the complex are unperturbed by the binding of KRAS" might be overstated. I suggest that the authors adopt a more nuanced approach in their conclusions. Ideally, this claim should be supported by experimental data, such as a cryoEM study of the KRAS/ Δ 155-BRAF complex. Investigating how Δ 155-BRAF, which more closely resembles mammalian BRAF, interacts with KRAS could provide informative insights.

We do not plan to undertake additional experimental studies to parse possible effects of insect vs mammalian cell expression on the RBD position. But we agree with the Reviewer that a more nuanced approach in our conclusions is reasonable. We have revised the discussion to state:

The structures described here show that engagement of BRAF by KRAS affects the position and interactions of the RBD domain, but does not directly disrupt core autoinhibitory interactions among the CRD domain, BRAF kinase domain, and the 14-3-3 and its recognition sites.

And we made the following edit in the Introduction, adding the underlined phrase:

The structures reveal a pre-activation intermediate, in which key autoinhibitory interactions of the CRD domain with the 14-3-3 dimer and BRAF kinase are unperturbed, despite engagement of KRAS.

And in the Abstract, adding the word “Core”:

Core autoinhibitory interactions in the complex are unperturbed by binding of KRAS and *in vitro* activation studies confirm that KRAS binding is insufficient to activate BRAF, absent membrane recruitment.

3. There are certain flaws in the proposed RAF activation model that need to be addressed, or alternatively, the text should be modified to reflect a more nuanced understanding.

As described below, we have modified the text to reflect a more nuanced understanding.

In my previous review, I requested experimental evidence for the dynamic conformational equilibrium between the "recruitment complex" and a hypothetical "open monomer" as a major activation step (step 2) in the proposed model. However, this remains unanswered. In response, the authors referred to previously published data on the S365A mutation in the BRAF/MEK/14-3-3 complex, which suggests a potential open monomer. This would indicate an equilibrium between the autoinhibited monomer and an open monomer, prior to step 1 in the proposed model, rather than an open monomer induced by RAS binding or post RAS binding (step 2). Thus, the second step is still speculative.

Additionally, I believe that a shift in elution position in preparative size exclusion chromatography alone is not strong evidence. Ideally, a technique that directly measures molecular mass, such as multi-angle light scattering coupled to SEC, analytical ultracentrifugation, or mass photometry, would be more suitable. However, I understand that performing these experiments is not requested at this time.

We agree and appreciate the Reviewer’s perspective. We have modified the text to indicate that the KRAS-bound open monomer state is hypothetical (we now refer to “... a hypothesized “open state” in which autoinhibitory interactions are released and pS365 is exposed.”)

In addition, we now conclude this section with an additional sentence:

This proposed open state of RAS-bound BRAF has not been directly observed, and additional studies will be required to vet this model.

Moreover, the authors concluded that "steric effects resulting from binding KRAS are not sufficient for the opening of the autoinhibited state." However, considering that the RBD position in the insect-cell expressed protein does not reflect the same position as in the mammalian expressed protein, this statement should be modified to reflect the system used in this study.

We revised this passage to read:

An important outstanding question is how “opening” of the autoinhibited complex is promoted in the membrane environment. It is clear from the present work that steric effects resulting from binding to KRAS are not sufficient to open the autoinhibited state, although they can contribute by disrupting the stabilizing interactions of the RBD with the 14-3-3 dimer seen in a prior structure ⁷.

Minor points:

- The blots in the main Figure 3 still lack molecular weight markers.

We have now added an indicator for the position of the 37 kDa molecular weight marker in Figure 3, which is the only marker in the cropped region. The full, uncropped blots with complete MW ladders are provided in the source data.

- It is still unclear whether the MST data represents replicates within a single experiment or the average of three independent experiments. To clarify, were the same dilution series measured three times, or was the dilution series prepared and measured three times? Additionally, the authors should comment on the large errors on the KD determinations and what that implies for the analysis.

The MST data are the average of three independent experiments. The dilution series were prepared and measured three times. We now explicitly state this in the methods section and in the supplementary figure legend:

Error bars represent standard deviations; experiments were performed three times, with independent dilutions for each (except for the CRD- kinase construct, for which no binding was observed in two independent experiments).

Additionally, the authors should comment on the large errors on the KD determinations and what that implies for the analysis.

We revised the text to state:

The standard deviations of these MST measurements are somewhat large, perhaps due to limitations in the highest RAF protein concentrations achievable. Nevertheless, the affinity values we obtain agree well with previous studies of KRAS binding to various BRAF and CRAF constructs (Table 1), including measurements of binding to the autoinhibited BRAF complex^{7,17,24}.

Reviewer #4 (Remarks to the Author):

In their manuscript “Cryo-EM structure of a RAS/RAF recruitment complex” Park et al describe the in vitro reconstructions of KRAS bound to intact BRAF in an autoinhibited state together with MEK1 and 14-3-3. Of interest, the authors observed a reduced resolution for the RAS-RAF RBD region indicating a relative mobility in that region, but can confirm interactions by crosslinking mass spectrometry (XL-MS).

The manuscript is well written and of great value for the readership of Nature Communications. The authors already gave detailed responses to the questions of previous reviewers. Hence, this reviewers’ comments will focus on the quality of the crosslinking data, which is also in line with the reviewers expertise.

The postulated conformational variability of RBD seems very exciting and novel to me, especially as it seems to be in contrast to the previously published structure by Fiesco in the context of membrane presence. Previous reviewers already suggested that the low resolution in the RAS-RAF RBD region prevents from reporting details on interaction and the authors responded that this region was too small to improve resolution. XL-MS data might be an optimal strategy useable to remodel the interaction surface and to confirm interaction. To that end I have several comments:

-Can you use your XL-MS data to do that? It seems from the supplementary dataset that no RBD to RAF or RAS XL was found and no link at all is reported on KRAS, but this is in contrast to Figure S4 where such links are plotted.

We did not identify any crosslinks in the RAS-RAF RBD region. The closest crosslink identified and shown in Figure S4 is an intraprotein linkage between K154 and K183 of BRAF. The only KRAS crosslink identified was an interprotein linkage between KRAS K147 and MEK1 K353. We did notice that a couple of the crosslinks reported in Figure S4 and in the supplementary annotated spectra were missing from the supplementary data file containing crosslink spectral matches. We have revised the supplementary dataset to include all confirmed crosslinks represented in Figure S4 and the annotated spectra.

-If you do not have sufficient links, the conclusions from XL-MS data would very much profit (if feasible) to be supplemented with XL data with a second linker that does not target Lys residues. E.g. SDA (Sulfo-NHS-Diazirine) is a heterobifunctional linker that would target all amino-acids after photoactivation increasing chances to confirm interactions within the investigated protein complex. Another option would be to couple carboxylic acids (i.e. D,E) to primary amines (i.e. K) using a coupling agent as DMTMM which would be a zero length linker enabling highest possible resolution from XL-MS data.

We agree that use of a linker that targets other amino acids could provide coverage of this region, and additional crosslinks to further define orientations of the KRAS/RBD. However, we are not pursuing this at present (in part because personnel changes in the lab have made it very difficult to undertake these studies).

-Since the BS3 links seem to be inconsistent to the CryoEM data suggesting flexibility (as also mentioned by Reviewer 2 in the previous revision), would it be possible to re-model the structure to obtain another fitting confirmation giving an idea to the other conformations existing? Tools like I-TASSER (iterative threading assembly refinement) or HADDOCK (high-ambiguity-driven protein-protein docking) have been used for this task previously, but there might be tons of others around as well.

It would be possible in principle to remodel the structure to conform to intersubunit crosslinks – for example we expect that the interprotein linkage between KRAS K147 and MEK1 K353 could be fit with a large rotation of the KRAS/RBD module, but given that this conformation would be defined by a single cross link we prefer not to go down this road.

-In Supplemental Figure 4, you show identified BS3 XLs: As mentioned before please check why this does not align with links reported in the supplemental data-set where I can only see interdomain links from 14-3-3 to BRAF and Intralinks but no other inter-domain link.

As stated above, we amended our supplemental dataset to include all confirmed crosslinks in Figure S4 and in the annotated spectra.

-In the same Figure the authors show which links do not fit to the maximum expected distance of 20Å. Can you elaborate in more detail how you confirmed those links are still confident (i.e. based on which FDR control was applied, why was a score difference of 15 for protein prospector chosen as sufficient – is this an empirical value?

In Protein Prospector, the crosslinked spectrum score difference is the difference in score of the crosslinked peptide match relative to the top-ranked assignment of a single peptide. Any score greater than 0 indicates that the crosslinked spectrum scored higher than the single peptide, which in theory should mean that there are fragment ions present from each linked peptide species. The higher the score difference, the more fragments from both linked species, the stronger the crosslinked spectrum match. Protein Prospector scoring of crosslinked peptides is more thoroughly explained in the reference below. We have included this reference in our manuscript. By manually inspecting a subset of the Protein Prospector search results, we found that implementing the preliminary filter of a score difference greater than 15 narrowed the search list to spectra that were more likely to be correct hits. This list was a manageable size for manual validation. We manually inspected each crosslinked species in the resulting list and removed any crosslinks that did not demonstrate sufficient fragmentation of each linked peptide. We have since reanalyzed this data with pLink2, which does allow for FDR filtering. At an FDR threshold of 1% (PSM level) with pLink2, we identified all linkages that passed our manual validation of the Protein Prospector data. We have included the results from both the pLink2 and Protein Prospector searches in the PRIDE submission (details below).

We considered the confidence of our CXMS data separately from that of our structural data. Most proteomics analyses rely on an FDR to demonstrate the data strength. This is certainly required when analyzing large datasets where manual validation is not an option. With this targeted structural analysis, however, we wanted to ensure that we had strong evidence of every crosslink we reported, and therefore further parsed our list to include only those were absolutely confident crosslink identifications through manually assessing the spectra. Our lists of unfiltered hits from the Protein Prospector search and the pLink2 search with a 1% FDR are included in the PRIDE submission.

Reference: Michael J Trnka, Peter R Baker, Philip J J Robinson, A L Burlingame, Robert J Chalkley. Matching Cross-linked Peptide Spectra: Only as Good as the Worse Identification. Mol Cell Proteomics. 2014 Feb; 13(2): 420–434. doi: 10.1074/mcp.M113.034009

-The authors describe, they manually validated their XLed spectra and provide a PDF showing all annotated spectra. I agree from my experience that those spectra look like confident matches to me as well. I am convinced though that a more detailed description of selection criteria/ chosen QC strategy /FDR threshold etc. would improve the quality of the manuscript. Especially the rather small database used for searching is potentially susceptible for false positive hits. Comparison to searches including some contaminants in the database as well, where no XL is expected to, could further improve confidence in the obtained results.

We have added a more detailed description concerning data filtering and validation to our methods section. We agree that the use of a small search database is more susceptible to false positives, which is why the manual validation step was necessary with our original approach. We have added to our PRIDE submission (details below) a pLink2 search against a target-decoy database containing complex member sequences as well as common contaminants. This approach does allow for FDR filtering and, at a 1% FDR at the PSM level, we identified all the hits that were manually validated with our original approach. Only one crosslink PSM out of a total of 1471 PSMs were mapped to a contaminant species. We agree that the comparison of the Protein Prospector and pLink2 searches, as well as the more detailed methods, improved the quality of the manuscript, and we thank the reviewer for this suggestion.

- Although the focus of this manuscript is on CryoEM, significant conclusions are driven from MS data as well which is why I would ask the authors to upload their XL-MS raw files, FASTA files, and ideally result files to a repository (e.g.

MassIVE <https://massive.ucsd.edu/ProteoSAFe/static/massive.jsp> ,
PRIDE <https://www.ebi.ac.uk/pride/archive/> , or others are commonly used in the proteomics community) to enable full transparency and reanalysis by others.

We have uploaded our raw files, fasta files, peaks lists, and search results to the PRIDE repository with Project Accession code *PXD042584*. Login details for the Reviewer Account have been provided in our cover letter to the editor to be shared with interested reviewers.

Minor

-Although this is likely not influenceable by the authors, it is noted here, that the given link to ProteinProspector does not work at the time of the review. The referenced paper (<https://www.ncbi.nlm.nih.gov/pmc/articles/PMC2596346/>) for ProteinProspector does not explain how this software performs XL searches but references to an ASMS paper instead (again with a broken link to the Prospector website). Hence, assessing details of functionality for that software was not possible for the reviewer.

We have provided a new link to the Protein Prospector website in the manuscript: <https://prospector.ucsf.edu/prospector/mshome.htm>. We have added a reference that explains the Protein Prospector crosslink search and validation in more detail.